# SAFB regulates hippocampal stem cell fate by targeting Drosha to destabilize *Nfib* mRNA

**Pascal Forcella[1†], Niklas Ifflander[1†], Chiara Rolando[1,2†], Elli-Anna Balta[1], Aikaterini Lampada[1], Claudio Giachino[1], Tanzila Mukhtar[1], Thomas Bock[3], Verdon Taylor[1]\***

[1]Department of Biomedicine, University of Basel, Basel, Switzerland; [2]Department of Biosciences, University of Milan, Milan, Italy; [3]Proteomics Core Facility, Biozentrum, University of Basel, Basel, Switzerland

**Abstract** Neural stem cells (NSCs) are multipotent and correct fate determination is crucial to guarantee brain formation and homeostasis. How NSCs are instructed to generate neuronal or glial progeny is not well understood. Here, we addressed how murine adult hippocampal NSC fate is regulated and described how scaffold attachment factor B (SAFB) blocks oligodendrocyte production to enable neuron generation. We found that SAFB prevents NSC expression of the transcription factor nuclear factor I/B (NFIB) by binding to sequences in the *Nfib* mRNA and enhancing Drosha-dependent cleavage of the transcripts. We show that increasing SAFB expression prevents oligodendrocyte production by multipotent adult NSCs, and conditional deletion of *Safb* increases NFIB expression and oligodendrocyte formation in the adult hippocampus. Our results provide novel insights into a mechanism that controls Drosha functions for selective regulation of NSC fate by modulating the post-transcriptional destabilization of *Nfib* mRNA in a lineage-specific manner.

**\*For correspondence:**
verdon.taylor@unibas.ch

[†]These authors contributed equally to this work

**Competing interest:** The authors declare that no competing interests exist.

## Editor's evaluation

The authors provide important information regarding the non-canonical functions of the ribonuclease Drosha in neural stem cell fate determination. They show convincing evidence that Drosha interacts with the Scaffold Attachment Factor B (SAFB) to prevent expression of the transcription factor NFIB, thereby preventing the formation of oligodendrocytes by neural stem cells. Overall their results provide new insight into molecular mechanisms that regulate NSC fate.

## Introduction

Neural stem cells (NSCs) are multipotent, generating the neurons and glia of the brain. NSCs produce neurons during development before switching to glial production and then disappearing from most regions of the brain. Therefore, NSC fate determination is precisely controlled. While neurogenesis is prominent at embryonic stages, in the adult vertebrate brain the generation of new neurons is predominantly restricted to two niches that maintain NSCs; the ventricular-subventricular zone (V-SVZ) of the lateral ventricles and the subgranular zone (SGZ) of the hippocampal dentate gyrus (DG) (*Obernier and Alvarez-Buylla, 2019*; *Gonçalves et al., 2016*). Whereas V-SVZ NSCs generate multiple neuron types, astrocytes and oligodendrocytes, DG NSCs predominantly generate glutamatergic granule neurons and, to a lesser extent, astrocytes but not oligodendrocytes (*Seri et al., 2004*; *Pilz et al., 2018*; *Bonzano et al., 2018*; *Bonaguidi et al., 2011*). Adult hippocampal neurogenesis plays a central role in memory formation, plasticity, and learning in rodents but has also been reported in

other species (*Eriksson et al., 1998*; *Spalding et al., 2013*; *Berg et al., 2019*; *Moreno-Jiménez et al., 2019*; *Gage, 2019*; *Boldrini et al., 2018*). Although still controversial, hippocampal neurogenesis persists in humans, and changes in neuron production might be linked to neurological diseases including Alzheimer's disease and epilepsy (*Moreno-Jiménez et al., 2019*; *Boldrini et al., 2018*; *Sorrells et al., 2018*; *Kempermann et al., 2018*; *Beckervordersandforth and Rolando, 2020*; *Tobin et al., 2019*). In rodents, adult NSC regulation has been studied extensively at the molecular and physiological levels, however, the importance of post-transcriptional regulation of gene expression in adult neurogenesis is less well understood (*Pilaz and Silver, 2015*; *Baser et al., 2019*).

The ribonuclease Drosha is a core component of the microRNA (miRNA) pathway and forms the miRNA Microprocessor with DGCR8 (*Han et al., 2004*; *Nguyen et al., 2015*). The Microprocessor cleaves primary-miRNA (pri-miRNA) stem loop hairpins (HPs) to generate precursor miRNA (pre-miRNA) in a specific pattern (*Kim et al., 2017*). Pre-miRNAs are further processed into mature miRNAs by Dicer before being directed to target mRNAs by the RNA-induced silencing complex. However, Drosha has direct, non-canonical post-transcriptional gene regulatory mechanisms beyond its function in miRNA biogenesis (*Rolando and Taylor, 2017*; *Chong et al., 2010*; *Lee and Shin, 2018*). Although miRNAs are important for terminal neuronal differentiation (*Yoo et al., 2011*), Drosha maintains the embryonic and hippocampal NSC pool independent of its miRNA biogenesis activity (*Knuckles et al., 2012*; *Rolando et al., 2016*). The proneural transcription factor Neurogenin2 specifies neuronal differentiation by NSCs and the expression of *Neurog2* gene is tightly controlled at the transcriptional level. At the post-transcriptional level, Drosha cleaves evolutionary conserved HPs in *Neurog2* mRNA transcripts inducing degradation and preventing translation (*Knuckles et al., 2012*). In addition to *Neurog2* transcripts, other mRNAs have been identified in NSCs that are cleaved by Drosha, including those of the transcription factors NeuroD1, NeuroD6, and NFIB (*Knuckles et al., 2012*; *Rolando et al., 2016*). NFIB is required and sufficient to promote glial fate specification by neural progenitors (*Deneen et al., 2006*). In the adult mouse hippocampus, Drosha destabilizes *Nfib* mRNAs preventing NFIB expression and thereby blocks DG NSCs from acquiring an oligodendrocytic fate (*Rolando et al., 2016*). *Nfib* mRNA contains HPs in the 5' untranslated region (UTR) and 3' UTR, which are both bound by Drosha and the 3' UTR HP is actively cleaved (*Rolando et al., 2016*). How Drosha binding to different mRNAs is regulated and cleavage specificity is achieved remains unclear.

Drosha's activity on target RNAs is regulated by its binding partners. Although DGCR8 and Drosha form the core of the Microprocessor, multiple Drosha-interacting proteins have been identified, underlining its functional diversity in RNA and transcriptional regulation (*Spadotto et al., 2018*; *Rouillard et al., 2016*). It remains elusive how Drosha-mediated cleavage of specific mRNAs is regulated during NSC maintenance and cell fate determination. We hypothesized that the specificity of Drosha activity in the regulation of NSC cell fate determination is precisely regulated by its interacting partners. Hitherto, the composition of the Drosha-containing protein complexes in NSC of the adult brain are unknown.

Using proteomic analyzes, we identified Drosha-interacting proteins in DG hippocampal NSCs and found that the RNA-binding protein (RBP), SAFB, regulates Drosha's activity in the destabilization of *Nfib* mRNA. SAFB is implicated in multiple cellular processes including cellular stress, DNA damage response, cell growth and apoptosis, and has been linked to miRNA biogenesis (*Altmeyer et al., 2013*; *Townson et al., 2000*; *Treiber et al., 2017*). Although SAFB is expressed in many tissues, its expression is particularly high in the brain (*Rivers et al., 2015*; *Townson et al., 2003*). Here, we demonstrate that SAFB levels are high in DG NSCs, and thereby block oligodendrocytic differentiation by regulating Drosha cleavage of the *Nfib* mRNA.

## Results
### Identification of Drosha-binding partners in DG NSCs

In order to examine the Drosha interactome in DG NSCs and identify proteins that potentially regulate canonical and non-canonical activities of Drosha in controlling NSC fate, we performed Drosha co-immunoprecipitation (co-IP) followed by tandem mass spectrometry (MS$^2$) from adult mouse DG NSCs (*Figure 1A* and *Figure 1—figure supplement 1A and B*). 165 proteins co-immunoprecipitated with Drosha (p<0.05, log$_2$ fold change ≥3, false discovery rate [FDR] ≤ 0.001, peptide count ≥4; *Figure 1B* and *Supplementary file 1*), the majority of which are RBPs (138; 84%) (*Gerstberger et al., 2014*;

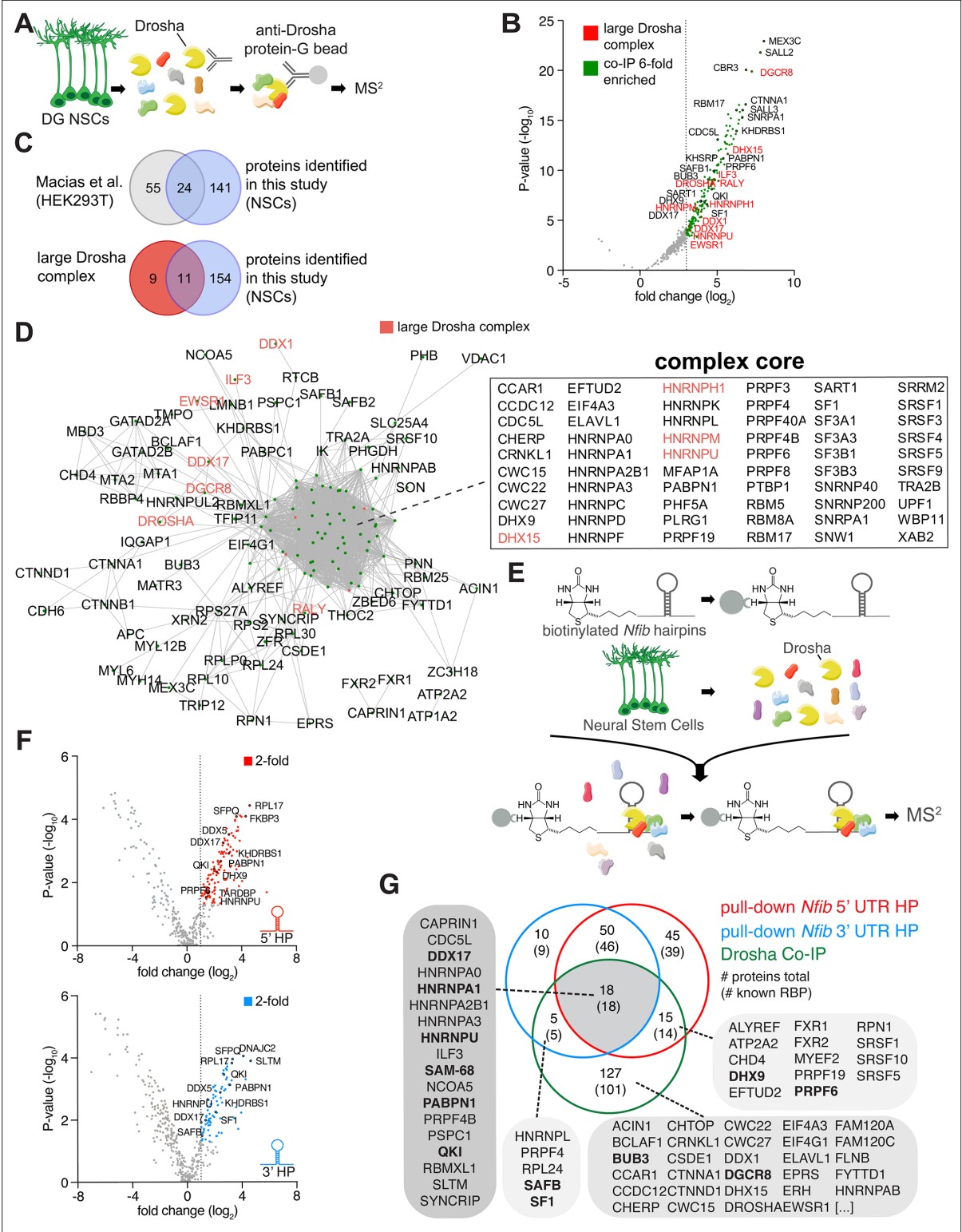

**Figure 1.** Identification of Drosha and *Nfib* mRNA with RNA-binding proteins in dentate gyrus (DG) neural stem cells (NSCs). (**A**) Scheme of the endogenous Drosha co-immunoprecipitation procedure from DG NSCs followed by tandem mass spectrometry (MS²) analysis. (**B**) Volcano plot of MS² quantified proteins displayed as fold change log₂ of Drosha co-precipitated proteins over control (x-axis) and p-value -log₁₀ (y-axis). Significantly enriched Drosha-associated proteins (green dots) were determined as having p-value < 0.05, log₂ fold change ≥3 and false discovery rate (FDR)≤0.001. Eleven proteins of the known 'large Drosha complex' (red) were also enriched in the co-precipitation with Drosha from DG NSCs. Selected novel proteins are also shown (black). For full list of MS² quantified proteins, see ***Supplementary file 1***. (**C**) Venn diagrams of pairwise comparisons of Drosha-

*Figure 1 continued on next page*

*Figure 1 continued*

interacting proteins identified in DG NSCs in this study and Drosha-interacting proteins in HEK293T cells and the CORUM large Drosha complex (*Rouillard et al., 2016*; *Macias et al., 2015*). (**D**) STRING network analysis of Drosha-interacting proteins in DG NSCs. Nodes are indicated as green dots; edges correspond to known interactions substantiated by experimental data. Only nodes with one or more edges are displayed, protein isoforms are shown collectively. The proteins in the densely packed core complex are listed alphabetically. (**E**) Scheme of the *Nfib* hairpin pull-down assay using biotinylated RNA probes as bait to capture binding proteins followed by precipitation with streptavidin-coupled beads and MS$^2$ analysis. (**F**) Volcano plots of MS$^2$ quantified *Nfib* 5′ untranslated region (UTR) hairpin (HP) (top - red) and *Nfib* 3′ UTR HP (bottom - blue) interacting proteins displayed as fold change log$_2$ (x-axis) versus p-value -log$_{10}$ (y-axis). Significant enriched proteins (red and blue dots) were defined as having p-value<0.05, log$_2$ fold change ≥1 and FDR≤0.1. Selected novel proteins are also shown (black). For full list of interacting proteins, see *Supplementary file 1*. (**G**) Venn diagram of comparisons between Drosha-interacting proteins in DG NSCs (green circle), *Nfib* 3′ UTR HP-binding proteins (blue circle), and *Nfib* 5′ UTR HP-binding proteins (red circle). The numbers indicate the total number of proteins, the numbers in brackets indicate how many of the proteins are known RNA-binding proteins (RBPs). Proteins selected for the functional assay are highlighted in the lists in bold type.

The online version of this article includes the following source data and figure supplement(s) for figure 1:

**Figure supplement 1.** Dentate gyrus (DG) neural stem cells (NSCs) contain large Drosha complexes.

**Figure supplement 1—source data 1.** Original data for *Figure 1—figure supplement 1B and F*.

**Figure supplement 1—source data 2.** Original data for *Figure 1—figure supplement 1B and F*.

**Figure supplement 2.** Process analysis of Drosha and *Nfib* mRNA-interacting proteins.

**Figure supplement 3.** *Nfib* mRNA interactome analysis uncovers regional binding preferences.

*Huang et al., 2018*). We compared the DG NSC Drosha interactome with results of a previous Drosha IP performed from human embryonic kidney cells (HEK293T) and found an overlap of 24 proteins (*Macias et al., 2015*). Comparison of the interactome with the 20 protein components of the large Drosha complex described previously (CORUM protein complexes dataset) revealed an overlap of 11 proteins (*Figure 1B and C*; *Rouillard et al., 2016*). Therefore, approximately 50% of the canonical large Drosha complex proteins identified in other cellular systems were identified as Drosha partner proteins in DG NSCs (*Figure 1B and C*). However, we identified over 100 novel Drosha partners. As expected, the canonical Drosha partner in the miRNA Microprocessor complex, DGCR8, was highly enriched in the DG NSC Drosha interactome (*Figure 1B*).

Process network analysis (MetaCore) of the Drosha-interacting proteins indicated that 56/165 proteins (34%, p=10$^{-61}$) are involved in the regulation of transcription and mRNA processing (*Figure 1—figure supplement 2A*). Conversely, only 2–7% of the Drosha-interacting proteins have been linked to other process networks and these include cell cycle, cell adhesion, and the cytoskeleton (p=10$^{-1}$–10$^{-3}$) (*Figure 1—figure supplement 2A*). In order to gain insights into the biological functions of the Drosha interactors, we performed Gene Ontology (GO) enrichment analysis of biological processes. The top GO term of the NSC Drosha interactome dataset was RNA splicing (GO:0008380) with >30-fold enrichment, followed by RNA processing (*Figure 1—figure supplement 2B*). As Drosha is primarily linked to RNA regulation, translation, and miRNA biogenesis, these findings support the specificity of the endogenous Drosha pull-down assay.

STRING functional protein association network analysis within the DG NSC Drosha interactome (considering only experimentally determined data and curated databases) (*Szklarczyk et al., 2019*) revealed one major complex indicating the close connectivity between interactors (*Figure 1D* and *Figure 1—figure supplement 1C and D*). Strikingly, many of the known Drosha interactors, including Drosha itself, were not positioned in the core of the complex suggesting additional mediators between Drosha and distant co-interactors. In summary, we identified novel Drosha-binding partners in adult DG NSCs, many of which are involved in transcriptional regulation and RNA biogenesis.

## Identification of interactors with *Nfib* 5′ UTR and 3′ UTR HP RNA sequences

We hypothesized that the specificity of Drosha activity in the regulation of NSC cell fate determination is precisely regulated by its interacting partners. With the compendium of Drosha-associated proteins at hand, we asked how these may affect Drosha activity in its role in direct destabilization of specific mRNAs to influence NSC fate. *Nfib* is required and sufficient to induce oligodendrocytic fate specification of NSCs and necessary for hippocampal development and myelination. NFIB expression is repressed in DG NSCs by Drosha-mediated post-transcriptional cleavage of its mRNA (*Rolando et al.,*

*2016*; *Deneen et al., 2006*). The 5′ UTR and 3′ UTR of the *Nfib* mRNA contain evolutionary conserved HPs that are both bound by Drosha (*Rolando et al., 2016*). To identify the proteins that potentially control *Nfib* mRNA stability, we performed pull-down experiments using the 5′ UTR and 3′ UTR HPs of the *Nfib* mRNA as a bait (*Rolando et al., 2016*).

We biotinylated RNA probes containing the *Nfib* 5′ UTR or 3′ UTR HP, including their respective flanking sequences and mixed these with protein lysates from DG NSCs. We precipitated the biotinylated probes and bound proteins with streptavidin beads and analyzed the associated proteins by MS$^2$ (*Figure 1E* and *Figure 1—figure supplement 1E*). As a proof of concept, we confirmed that Drosha was bound to and was precipitated with both the *Nfib* 5′ UTR and 3′ UTR HP probes (*Figure 1—figure supplement 1F*). We identified 128 proteins bound to the *Nfib* 5′ UTR, and 83 proteins bound to the 3′ UTR HP probes ($p<0.05$, log$_2$ fold change ≥1, FDR ≤ 0.1, peptide count ≥4; *Figure 1G*). As a comparison and negative control for enrichment in these experiments, we used a biotinylated RNA probe of similar length to the *Nfib* HP probes but corresponding to the proximal 3′ UTR of the androgen receptor (AR) mRNA as a bait (*Figure 1—figure supplement 1F* and *Supplementary file 2*). Proteins that bound to the AR control probe and were not significantly enriched in the *Nfib* 5′ UTR and 3′ UTR HP probe pull-downs were considered as general RNA-binding proteins or background.

The majority of the *Nfib* HP interactors (117/128: 5′ UTR HP bound proteins, and 78/83: 3′ UTR HP bound proteins) are known RBPs (*Gerstberger et al., 2014*; *Huang et al., 2018*; *Figure 1G*). MetaCore and GO term analysis of process networks revealed that 22% and 27% of the *Nfib* 5′ UTR HP- and 3′ UTR HP-binding proteins, respectively, are known to be involved in transcription (*Figure 1—figure supplement 2C–F*). Strikingly, translation initiation (21% of total, p-value $10^{-23}$) and elongation (22% of total, p-value $10^{-21}$) were strongly enriched in the 5′ UTR HP-interacting protein set but were far less relevant for the 3′ UTR HP-interacting protein sets (both 9% of total, p-value $10^{-5}$ and $10^{-4}$) (*Figure 1—figure supplement 2C–F*). Similarly, STRING network analysis of 5′ UTR and 3′ UTR HP-binding proteins resulted in comparable findings (*Figure 1—figure supplement 2G and H*). While both interaction datasets include many heterogeneous nuclear ribonucleoproteins (hnRNPs), only the 5′ UTR HP-interacting proteins contained an additional complex that comprised many ribosomal proteins (*Figure 1—figure supplement 2G*).

## *Nfib*-interacting RBPs bind the HP flanking regions

The binding of Drosha to cleavage sites in target mRNAs is not understood (*Rolando et al., 2016*). We assessed whether the proteins that associated with the *Nfib* mRNA recognize and directly bind the sequences that form the 5′ UTR and 3′ UTR HPs or potentially the sequences flanking these HPs. Therefore, we generated biotinylated hybrid RNA probes that contained the respective flanking sequences of the *Nfib* mRNA 5′ UTR and 3′ UTR HPs, but where the HP-forming nucleotides themselves had been replaced by a none HP-forming RNA sequence from the AR mRNA (AR RNA/*Nfib* 5′ UTR flanking and AR RNA/*Nfib* 3′ UTR flanking) (*Figure 1—figure supplement 1E* and *Figure 1—figure supplement 3A and C*). Hnrnpa2b1, Purb, Vcp, Pura, and Rbm25 were enriched in the pull-downs of the *Nfib* 5′ UTR HP probe compared to its corresponding control AR hybrid probe, and RBMX was the only protein that selectively interacted with the *Nfib* 3′ UTR HP compared to its corresponding control AR hybrid probe (*Figure 1—figure supplement 3A and C* and *Supplementary file 2*). Thus, the majority of the RBPs in the *Nfib* 5′ UTR HP and 3′ UTR HP interaction datasets selectively bind to the regions immediately flanking the HP-forming sequences in the 5′ and 3′ UTRs of the *Nfib* mRNA. GO biological process analysis of the proteins associated with the HP flanking regions revealed that translation (GO:0006412) was 10-fold enriched in the 5′ UTR HP flanking sequence-associated protein dataset compared to the random control prediction, whereas regulation of gene expression and RNA processing were most enriched in the 3′ UTR HP flanking region binding protein dataset (*Figure 1—figure supplement 3B and D*).

Previously, we showed that the 3′ UTR HP region of the *Nfib* mRNA is cleaved by Drosha, and this contributes to the destabilization of the RNA and blockade of NFIB expression (*Rolando et al., 2016*). Comparison of the *Nfib* 5′ UTR and 3′ UTR HP-interacting proteins revealed 18 proteins that bind preferentially to the *Nfib* 3′ UTR HP region, with Hnrnpl, SAFB, Sfpq, and Nono showing the highest enrichment ($p<0.05$, log$_2$ fold change ≥1, FDR ≤ 0.1, peptide count ≥4; *Figure 1—figure supplement 3E* blue dots and *Supplementary file 2*). GO analysis of the *Nfib* 3′ UTR HP-specific

interacting proteins showed significant enrichment in proteins associated with negative regulation of mRNA metabolic processes (GO:1903312) (*Figure 1—figure supplement 3F*).

We hypothesized that the proteins interacting with Drosha and *Nfib* HPs could facilitate Drosha-mediated direct cleavage of *Nfib* mRNAs. We compared the lists of Drosha-, *Nfib* 3'-, and 5' UTR-interacting proteins (*Figure 1G* and *Supplementary file 3*) and identified 18 RBPs, included the known Drosha-binding proteins and Hnrnp family members Hnrnpa1, Hnrnpa2b1, and Hnrnpu, as putative modulators of Drosha activity in *Nfib* mRNA processing (*Rouillard et al., 2016*; *Gerstberger et al., 2014*; *Huang et al., 2018*; *Macias et al., 2015*). We identified 5 RBPs that interacted with Drosha and the 3' UTR HP but not the 5' UTR HP region of *Nfib* and 14 RBPs that bound Drosha and the 5' UTR HP but not the 3' UTR HP region of the *Nfib* mRNA. These data supported the hypothesis that different Drosha-containing complexes interacted selectively with specific RNA sequences in the *Nfib* mRNA. The majority of the Drosha-binding partners (127/165 identified proteins in the Drosha IP), including DGCR8 as the partner of Drosha in miRNA biogenesis, did not interact with the *Nfib* 3' UTR HP or 5' UTR HP regions. This supports that the Drosha protein complex that regulates *Nfib* mRNA stability is distinct from the core miRNA Microprocessor complex, and that different protein complexes are likely involved in the different Drosha canonical and non-canonical pathways.

## Identification of modulators of non-canonical Drosha activity

To elucidate regulatory functions of the *Nfib* mRNA-interacting RBPs in Drosha-mediated control of *Nfib* mRNA stability, we developed a Drosha cleavage activity reporter system in DG NSCs. We generated stable DG NSC lines carrying floxed *Drosha* alleles (*Drosha^{fl/fl}*) for conditional ablation in vitro and a destabilized green fluorescent protein (EGFPd2) driven from a doxycycline-regulated expression cassette (*Li et al., 1998*). One *Drosha^{fl/fl}* DG NSC line stably expressed the Drosha insensitive control EGFPd2 reporter (Tet-on ctrl) (*Figure 2A*). To report endogenous Drosha-mediated destabilization of *Nfib* mRNA, we generated a *Drosha^{fl/fl}* DG NSC line where the *Nfib* 3' UTR HP sequence was inserted downstream of the doxycycline-regulated EGFPd2 coding region (Tet-on 3' UTR HP) (*Figure 2A and B*). Importantly, both Tet-on ctrl and Tet-on 3' UTR HP NSC lines retained stem cell properties including the capacity to generate neurons upon differentiation (*Figure 2B*).

EGFPd2 (referred to hereafter as GFP) expression was induced to submaximal levels by tittered administration of doxycycline in order to quantify both decreases and increases in GFP expression (*Figures 2C, D, and 3B*, left). We assessed the ability of these inducible DG NSC lines to report Drosha cleavage of the *Nfib* mRNA. We conditionally ablated *Drosha* (*Drosha* cKO) from the Tet-on ctrl and Tet-on 3' UTR HP reporter lines by transient expression of Cre-recombinase (Cre-IRES-Tom) (*Figure 2A, E, and F*) followed by doxycycline induction. We quantified GFP expression by the *Drosha* cKO (Tomato^+) and the *Drosha* wild type (WT) NSCs (Tomato^-) in the Tet-on ctrl and Tet-on 3' UTR NSC cultures by flow cytometry after 48 hr (*Rolando et al., 2016*; *Figure 2E*). *Drosha* cKO Tet-on 3' UTR NSCs (Tomato^+) showed an increase in GFP expression compared to Tet-on 3' UTR NSCs with intact Drosha alleles (Tomato^-). Conversely, *Drosha* cKO did not affect GFP expression in the Tet-on ctrl NSC line (*Figure 2G and H*). Therefore, the presence of the 3' UTR HP of the *Nfib* mRNA in the Tet-on 3' UTR HP reporter conveyed Drosha sensitivity, enabling quantification of Drosha cleavage of target mRNAs to screening the effects of Drosha partners.

## Drosha and *Nfib* mRNA-interacting proteins are novel regulators of *Nfib* HP processing

We hypothesized that common Drosha- and *Nfib* mRNA-interacting proteins modulate Drosha activity toward the *Nfib* mRNA. Therefore, we selected RBPs from the Drosha-interacting protein dataset which also bound the *Nfib* 3' UTR HP probe in the pull-down assays and performed gain-of-function analysis in the Tet-on and Tet-on 3' UTR HP reporter DG NSC lines (*Figure 3A*). We transiently expressed Bub3, Ddx5, Ddx17, DGCR8, Dhx9, Fus, Hnrnpa1, Hnrnpu, Khsrp, Pabpn1, Prpf6, Qki, SAFB, Sam68, Sart1, Sf1, Tdp43, and Trim9 (CAG::Rbp-IRES-Cfp) or CAG::IRES-Cfp as a control in Tet-on and Tet-on 3' UTR HP reporter NSC lines and quantified the effects of each RBP on GFP expression at the protein and RNA levels after doxycycline induction. We performed single-cell quantification of GFP levels by flow cytometric analysis and comparison of CFP^+ and CFP^- NSCs in Tet-on and Tet-on 3' UTR HP reporter lines, and at the mRNA level by quantitative reverse transcriptase PCR (RT-qPCR) for EGFPd2 mRNA on sorted CFP^+ and CFP^- NSCs from each transfection (*Figure 3A and B*).

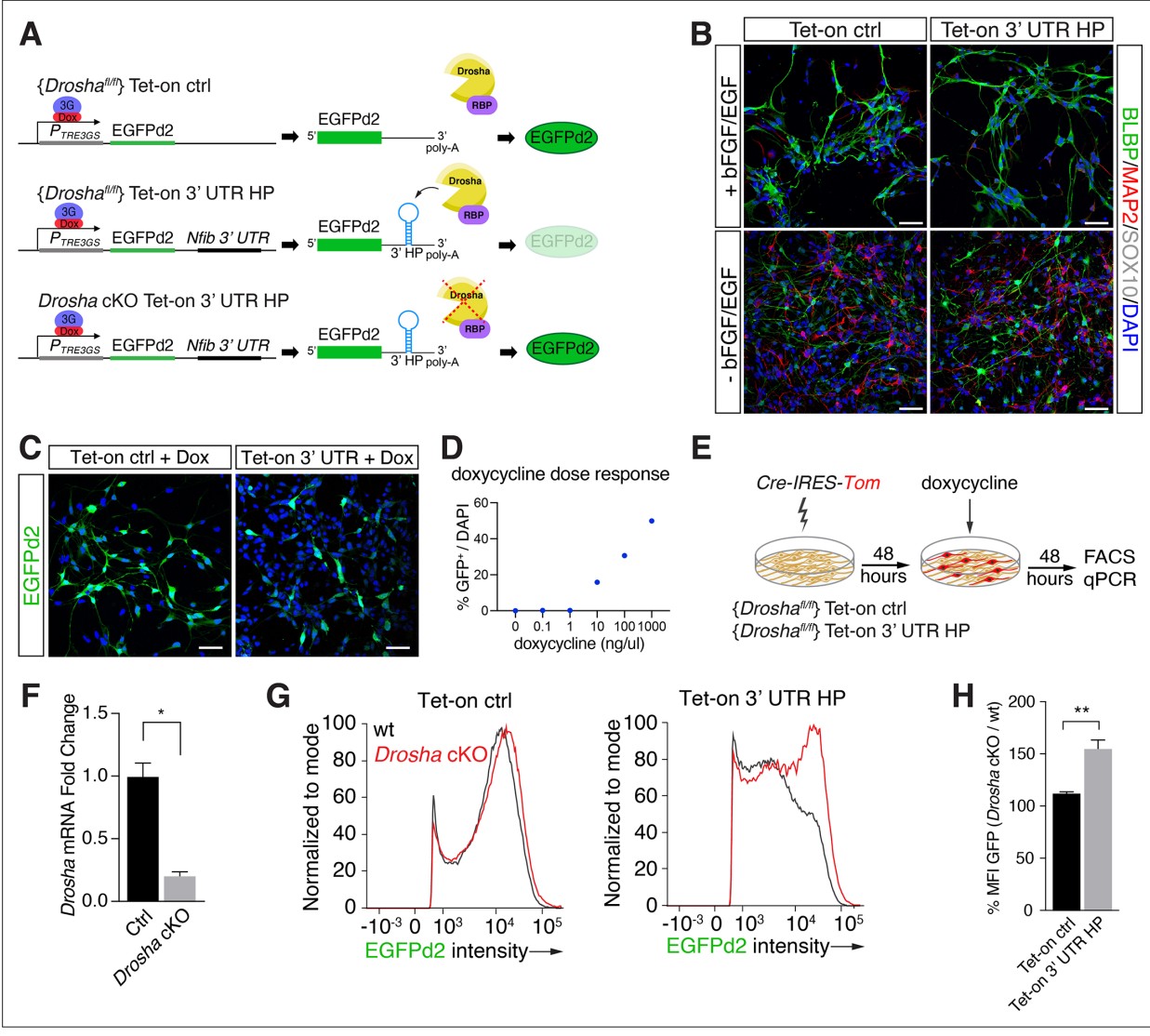

**Figure 2.** Conditional dentate gyrus (DG) neural stem cells (NSCs) report Drosha processing of *Nfib* constructs. (**A**) Scheme of the experimental paradigm using the Tet-on reporter lines to examine the effects of the *Nfib* 3' untranslated region (UTR) hairpin (HP) (composed of *Nfib* 3' UTR HP inserted into the UTR downstream of EGFPd2 *cDNA*) on expression. Stable floxed *Drosha* (*Drosha*^fl/fl^) DG NSCs lines carrying doxycycline inducible Tet-on ctrl, or Tet-on 3' UTR HP constructs were generated and selected. Drosha/RNA-binding protein (RBP) complexes regulate stability of the reporter RNA and EGFPd2 expression levels. Deletion of Drosha stabilizes the Tet-on 3' UTR HP construct mRNA and EGFPd2 expression. (**B**) Characterization of the Tet-on ctrl and *Nfib* 3' UTR HP (Tet-on 3' UTR) expressing DG NSCs under expansion (+bFGF/EGF) and differentiation (-bFGF/EGF) culture conditions. Immunohistochemistry for the progenitor marker BLBP, neuronal protein MAP2, and oligodendrocyte protein SOX10. Scale bars 50 μm. (**C**) Expression of EGFPd2 by doxycycline-induced (48 hr) Tet-on ctrl and Tet-on *Nfib* 3'UTR HP DG NSC line under *Drosha*^fl/fl^ conditions. Scale bar 50 μm. (**D**) Doxycycline dose-response curve of Tet-on ctrl DG NSC line. Quantification of EGFPd2+ (GFP+) cells over total cells (DAPI). (**E**) Experimental paradigm for *Drosha* conditional deletion (*Drosha* cKO) experiments from Tet-on ctrl and Tet-on 3' UTR HP DG NSCs followed by quantitative FACS analysis for EGFPd2 expression and quantitative reverse transcriptase PCR (RT-qPCR). (**F**) RT-qPCR quantification of Drosha mRNA levels before (Ctrl) and after *Drosha* cKO from Tet-on 3' UTR HP DG NSCs. n=4, two-tailed Mann-Whitney test: *p<0.05. Error bars SEM. (**G**) FACS analysis of EGFPd2 fluorescence by Tet-on ctrl and Tet-on 3' UTR HP DG NSCs after doxycycline induction (48 hr). EGFPd2 intensity (x-axis) versus cell number normalized to mode (y-axis) of Tet-on ctrl and Tet-on 3' UTR HP DG NSC lines before *Drosha* cKO (WT: black line) and after *Drosha* cKO (red line). Note the recovery of high EGFPd2-expressing cells (intensity >10^4) in the *Drosha* cKO Tet-on 3' UTR HP DG NSC (red line) compared to the same cells before Drosha deletion (WT: black line). (**H**) Quantification of median fluorescence intensity of EGFPd2 (GFP) from *Drosha* cKO over WT in Tet-on ctrl and Tet-on 3' UTR HP lines. n=5, two-tailed Mann-Whitney test: **p<0.01. Error bars SEM.

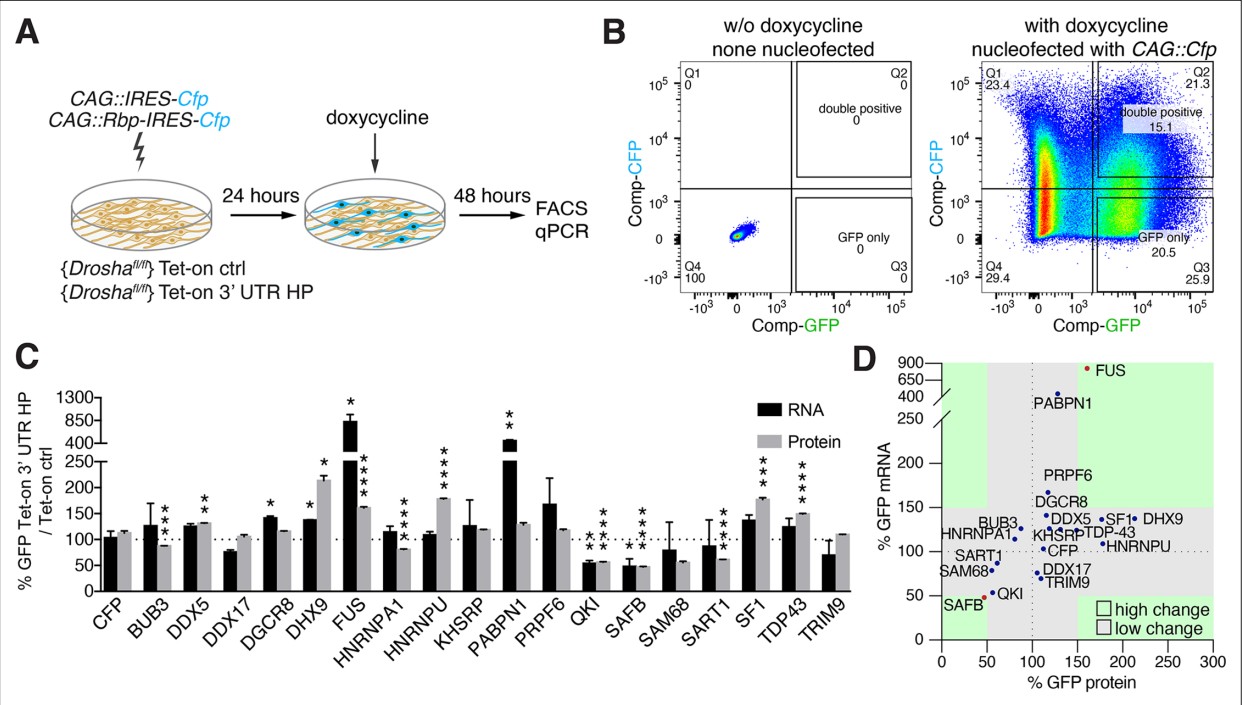

**Figure 3.** Overexpression of Drosha interactors affects cleavage of *Nfib* mRNA. (**A**) Scheme of the experimental setup to screen the effects of RNA-binding proteins (RBPs) on the expression of Tet-on ctrl and Tet-on 3' untranslated region (UTR) hairpin (HP) constructs in dentate gyrus (DG) neural stem cells (NSCs) by FACS and quantitative reverse transcriptase PCR (RT-qPCR) analysis. RBPs (CAG::Rbp-IRES-Cfp) were expressed in Tet-on ctrl and Tet-on 3' UTR HP DG NSCs by nucleofection and the levels of EGFPd2 (GFP) protein and mRNA compared to the expression by NSCs expressing a CFP control construct (CAG::IRES-Cfp) after 48 hr of doxycycline induction. (**B**) Representative FACS plot from flow cytometric analyzes of Tet-on ctrl DG NSCs with or without nucleofection with the control (CAG::IRES-Cfp) expression vector, with or without doxycycline induction (48 hr). Untransfected and uninduced Tet-on ctrl DG NSCs do not show EGFPd2 (GFP) or CFP expression compared to doxycycline-induced and CAG::IRES-Cfp nucleofected Tet-on ctrl DG NSCs. (**C**) Analysis of RBP effects on Tet-on 3' UTR HP construct expression in doxycycline-induced Tet-on 3' UTR HP DG NSCs. Nucleofected cells were sorted by FACS, gating on the CFP⁺ cells. EGFPd2 (GFP) levels of CFP⁺ cells were quantified by flow cytometry and RNA isolated for RT-qPCR analysis. Quantification of relative EGFPd2 (GFP) mRNA (RT-qPCR) and protein levels (flow cytometry) following overexpression of RPBs by nucleofection in Tet-on 3' UTR HP DG NSCs compared to control vector (CAG::IRES-Cfp) transfected cells. To eliminate effects of the RBPs on transcription, RNA stability or translation not linked to the *Nfib* 3' UTR HP, the changes in expression were calculated as the differences in EGFPd2 levels in Tet-on 3' UTR HP DG NSCs and Tet-on ctrl DG NSCs (%GFP Tet-on 3' UTR HP/Tet-on ctrl). Black dotted line indicates no change. Error bars SEM. (**D**) Summary diagram of RBP effects on Tet-on 3' UTR HP relative to Tet-on ctrl construct expression in DG NSCs. Relative EGFPd2 (GFP) protein levels (fluorescence intensity by FACS, x-axis) and mRNA levels (RT-qPCR, y-axis). Green areas represent changes of ±>50% of Tet-on 3' UTR HP relative to Tet-on ctrl construct.

In order to identify RBPs that affected reporter expression in an *Nfib* 3' UTR HP-dependent fashion, we normalized the changes in GFP expression induced by the overexpressed RBP in the Tet-on 3' UTR HP DG NSCs to the changes induced in the Tet-on ctrl DG NSCs. Among the 18 RBPs tested, Dhx9, Sf1, Hnrnpu, TDP-43, and FUS caused increases in GFP expression in an *Nfib* 3' UTR HP-dependent fashion. Conversely, SAFB, Qki, Sam68, and Sart1 induced robust reductions in GFP protein expression (*Figure 3C and D*). Fus and SAFB also induced corresponding changes in the EGFPd2 mRNA levels (FUS 824%, SAFB 48%) suggesting that they act by regulation of mRNA stability (*Figure 3D*). As SAFB strongly reduced GFP mRNA and protein expression in these assays, SAFB was a prime candidate for regulating Drosha catalytic activity in the regulation of *Nfib* mRNA stability (*Figure 3D*).

## SAFB regulates *Nfib* mRNA stability via Drosha at the 3' UTR HP

SAFB showed a strong binding preference toward the *Nfib* 3' UTR HP compared to the *Nfib* 5' UTR HP, which correlates with the cleavage activity of *Nfib* mRNA by Drosha (*Figure 1—figure supplement 3E*). Therefore, in order to compare the effects of SAFB on the *Nfib* 5' UTR HP versus the *Nfib* 3' UTR HP in this assay, we generated a DG NSC line (Tet-on 5' UTR HP), expressing a Tet-on *Nfib* construct where the 5' UTR HP replaced the *Nfib* 3' UTR HP downstream of the EGFPd2 expression cassette (*Figure 4A and B* and *Figure 4—figure supplement 1A and B*). Tet-on 5' UTR HP NSCs

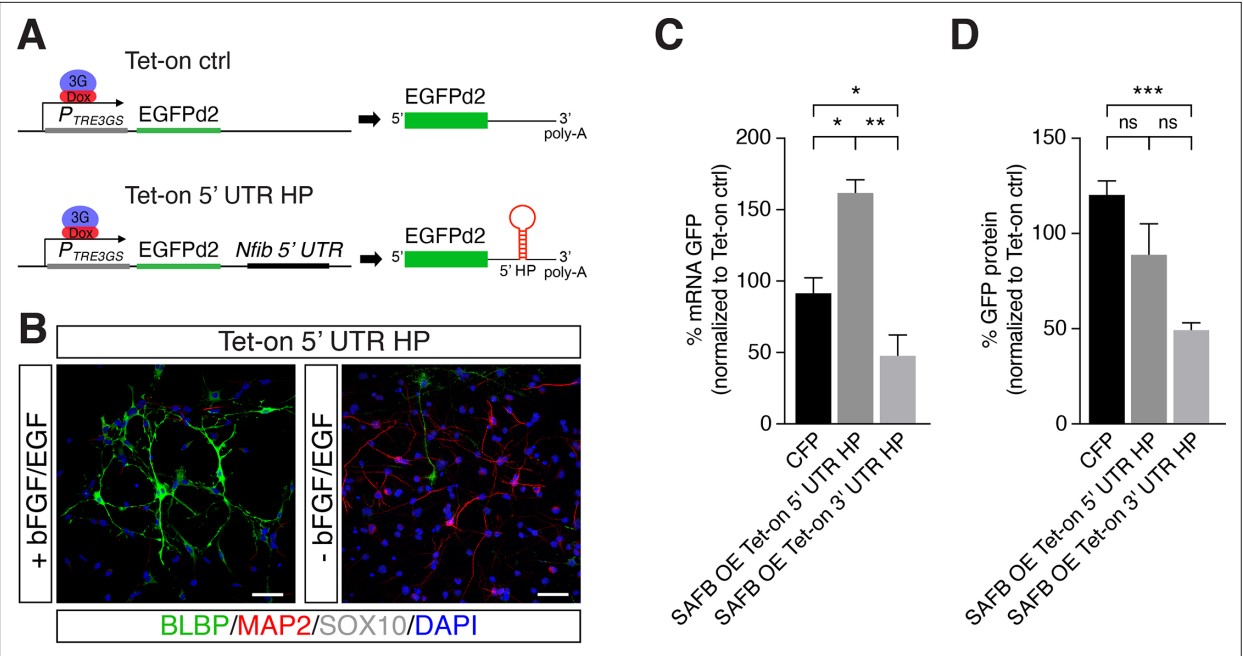

**Figure 4.** SAFB overexpression modulates *Nfib* reporter expression. (**A**) Scheme of the constructs used for generating Tet-on ctrl and Tet-on 5' untranslated region (UTR) hairpin (HP) dentate gyrus (DG) neural stem cells (NSCs). (**B**) Characterization of the *Nfib* 5' UTR HP expressing DG NSCs under expansion (+bFGF/EGF) and differentiation (-bFGF/EGF) culture conditions. Immunohistochemistry for the progenitor marker BLBP, neuronal protein MAP2, and oligodendrocyte protein SOX10. Scale bar 50 µm. (**C**) Quantitative reverse transcriptase PCR (RT-qPCR) analysis of EGFPd2 (GFP) mRNA levels of CFP expressing and SAFB overexpressing (SAFB OE) Tet-on 3' UTR HP and Tet-on 5' UTR HP DG NSCs (x-axis). Percent mean EGFPd2 (GFP) mRNA expression by Tet-on 3' UTR HP and Tet-on 5' UTR HP DG NSCs normalized to expression by Tet-on ctrl DG NSCs (y-axis). n=3; one-way ANOVA with Holm-Sidak's test: *p<0.05, **p<0.01. Error bars SEM. (**D**) FACS analysis of EGFPd2 (GFP) protein fluorescence of CFP expressing and SAFB overexpressing (SAFB OE) Tet-on 3' UTR HP and Tet-on 5' UTR HP DG NSCs (x-axis). Percent median EGFPd2 (GFP) protein fluorescence intensity of Tet-on 3' UTR HP and Tet-on 5' UTR HP DG NSCs normalized to fluorescence intensity of Tet-on ctrl DG (y-axis). n=6; one-way ANOVA with Holm-Sidak's test: ***p<0.001, ns - not significant. Error bars SEM.

The online version of this article includes the following source data and figure supplement(s) for figure 4:

**Figure supplement 1.** Dentate gyrus (DG) neural stem cell (NSC) *Nfib* untranslated region (UTR) reporter lines retain NSC properties.

**Figure supplement 1—source data 1.** Original data for *Figure 4—figure supplement 1B*.

**Figure supplement 1—source data 2.** Original data for *Figure 4—figure supplement 1B*.

like the Tet-on 3' UTR HP NSCs also retained stem cell properties including the capacity to generate neurons (MAP2[+]) and astrocytes (S100β[+]), but not oligodendrocytes (SOX10[+]) upon differentiation (*Figure 4B*, *Figure 4—figure supplement 1A*, and *Figure 2B*).

We expressed SAFB (CAG::Safb-IRES-Cfp) and CFP alone (CAG::IRES-Cfp) in Tet-on ctrl, Tet-on 5' UTR HP, and Tet-on 3' UTR HP DG NSCs and compared the levels of GFP protein and EGFPd2 mRNA in CFP[+] and CFP[-] cells after doxycycline induction (*Figure 3A*). SAFB overexpression significantly reduced EGFPd2 mRNA expression in Tet-on 3' UTR HP DG NSCs but not in 5' UTR HP DG NSCs (*Figure 4C*). By contrast, SAFB overexpression increased EGFPd2 mRNA in Tet-on 5' UTR HP DG NSCs indicating key differences in SAFB effects on the two HPs in these assays (*Figure 4C*). Similarly, GFP protein intensity was also reduced in the Tet-on 3' UTR HP DG NSCs following SAFB expression (*Figure 4D*). These data showed that SAFB can regulate *Nfib* mRNA stability through the 3' UTR HP.

We addressed whether SAFB and Drosha cooperate to regulate endogenous *Nfib* mRNA levels. First, we investigated whether Drosha and SAFB physically interact in DG NSCs and performed co-IP assays. We immunoprecipitated endogenous Drosha from DG NSCs with an anti-Drosha antibody and detected endogenous SAFB in the precipitate but not PKC-α. We used PKC-α as a negative control as it has not been reported to interact with Drosha and was not identified in the Drosha IP MS[2] analyzes (*Figure 5A* and *Figure 5—figure supplement 1A–C*). In addition, we performed SAFB co-IP assays from DG NSCs with anti-SAFB antibodies and detected Drosha in the precipitate confirming the binding of SAFB to Drosha (*Figure 5—figure supplement 1D–F*). We then addressed whether

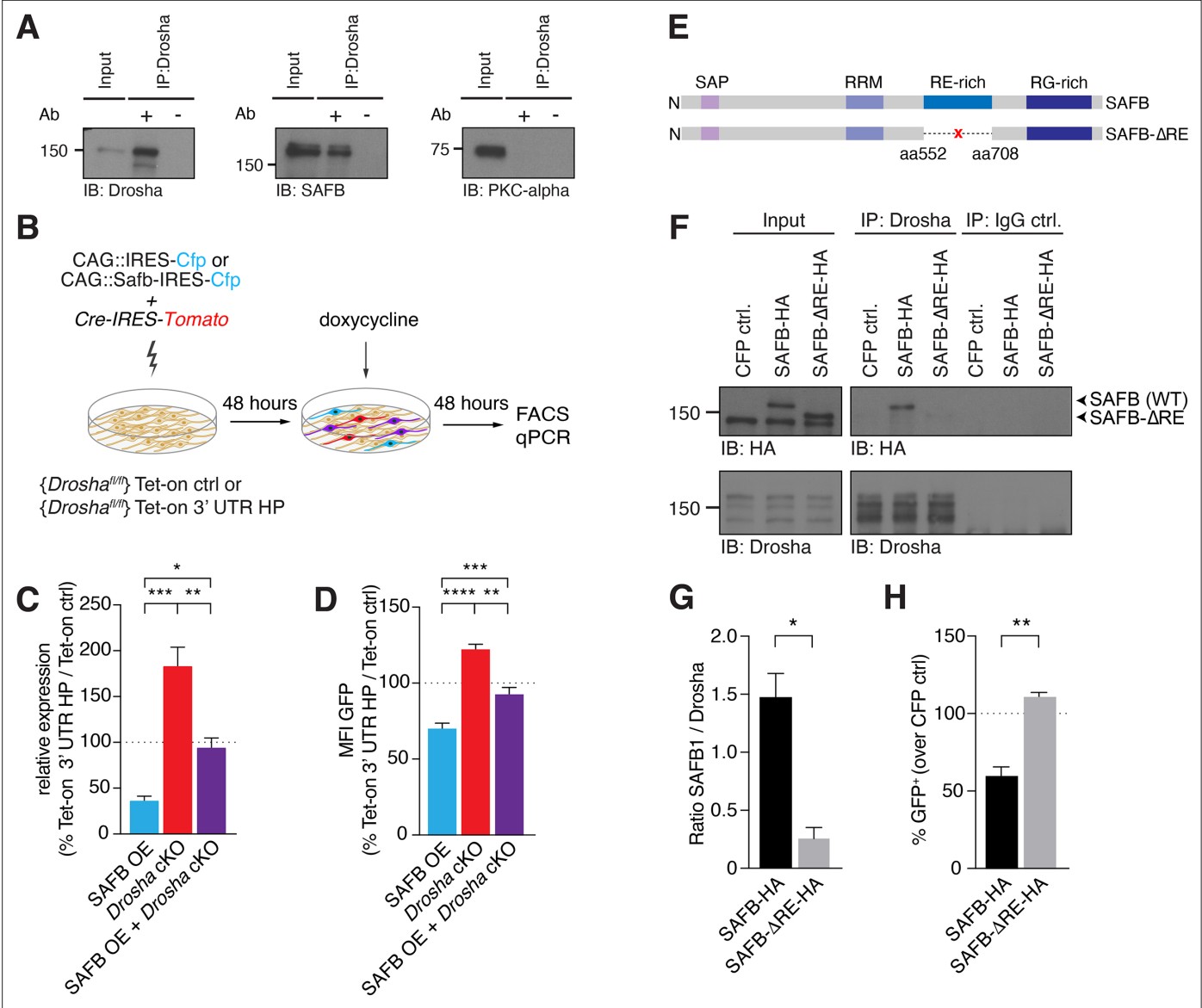

**Figure 5.** SAFB-mediated reduction in *Nfib* expression depends on Drosha activity. (**A**) Immunoblot validation of Drosha-SAFB interaction. Drosha immunoprecipitation (IP) shows enrichment of Drosha (13% of input) and co-IP of SAFB. Input: 2.5% of total lysate. PKC-α was used as negative control for the co-IP. (**B**) Scheme of the experimental setup for analysis of the effects of SAFB overexpression (OE) (CAG::Safb-IRES-Cfp) on Tet-on ctrl and Tet-on 3' untranslated region (UTR) HP expression in floxed *Drosha* (*Drosha*^fl/fl^) dentate gyrus (DG) neural stem cells (NSCs) by FACS and quantitative reverse transcriptase PCR (RT-qPCR) analysis. (**C**) RT-qPCR analysis of EGFPd2 (GFP) mRNA levels in Tet-on 3' UTR HP DG NSCs. Quantification of mean EGFPd2 (GFP) mRNA expression in Tet-on 3' UTR HP DG NSCs after *Drosha* cKO and OE of SAFB with/without *Drosha* cKO relative to Tet-on ctrl DG NSCs. 100% line indicates no difference in expression. n=3, one-way ANOVA with Holm-Sidak's test: *p<0.05, **p<0.01, ***p<0.001. Error bars SEM. (**D**) FACS analysis of EGFPd2 (GFP) protein fluorescence in Tet-on 3' UTR HP DG NSCs. Quantification of median EGFPd2 intensity (MFI GFP) of Tet-on 3' UTR HP DG NSCs after *Drosha* cKO and OE of SAFB with/without *Drosha* cKO relative to Tet-on ctrl DG NSCs. 100% line indicates no difference in expression. n=5, one-way ANOVA with Holm-Sidak's test: *p<0.05, **p<0.01, ****p<0.0001. Error bars SEM. (E) Schematic representation of the SAFB and SAFB-ΔRE mutant lacking amino acids 552–708 encoding the RE domain. (**F**) Immunoblot (IB) analysis of SAFB-HA (wild type [WT]), SAFB-ΔRE-HA and Drosha in the Drosha IP and IgG isotype control IP from transfected N2A cells. Input: 2.5% of total lysate. (G) Quantification of binding capacity of Drosha to SAFB and SAFB-ΔRE. Binding was quantified as the levels of SAFB-HA or SAFB-ΔRE-HA relative to Drosha in the respective co-IP measured by densitometry of immunoblots. n=3, two-tailed t-test with Welch's correction: *p<0.05. Error bars SEM. (H) FACS analysis of EGFPd2 (GFP) protein fluorescence in Tet-on 3' UTR HP DG NSCs following SAFB and SAFB-ΔRE OE. Percentage of median EGFPd2 (GFP) intensity over CAG::IRES-Cfp control transfection. 100% line indicates no difference in expression. n=4, two-tailed Mann-Whitney test: **p<0.01. Error bars SEM.

The online version of this article includes the following source data and figure supplement(s) for figure 5:

**Source data 1.** Original data for *Figure 5* and *Figure 5—figure supplement 1*.

*Figure 5 continued on next page*

*Figure 5 continued*

**Source data 2.** Original data for *Figure 5* and *Figure 5—figure supplement 1*.

**Figure supplement 1.** SAFB and Drosha are able to form a protein complex.

the Drosha-SAFB interactions required RNA and performed SAFB co-IP in the presence of RNase. Drosha co-precipitated with SAFB in the presence of RNase suggesting that an RNA molecule was not required for their interaction (*Figure 5—figure supplement 1D–F*).

In order to address whether SAFB and Drosha cooperate at the *Nfib* 3′ UTR HP to regulate *Nfib* mRNA levels, we analyzed the effects of increasing SAFB expression on Tet-on 3′ UTR HP reporter activity in the presence and absence of Drosha (*Figure 5B*). In the presence of Drosha, SAFB over-expression reduced GFP expression from the doxycycline-induced Tet-on 3′ UTR HP reporter at the RNA (RT-qPCR) and protein levels (FACS), compared to the Tet-on GFP reporter (*Figure 5B–D*). We addressed whether the SAFB effects on Tet-on 3′ UTR HP reporter expression depended on Drosha activity by expressing SAFB and simultaneously conditionally deleting *Drosha* (*Drosha* cKO). While *Drosha* cKO increased GFP expression from the Tet-on 3′ UTR HP reporter at the mRNA and protein levels, simultaneous *Drosha* cKO and overexpression of SAFB reversed the GFP reduction in Tet-on 3′ UTR HP reporter expression at both the RNA and protein levels (*Figure 5B–D*). Therefore, Drosha and SAFB work together to regulate the expression of the Tet-on 3′ UTR HP reporter in NSCs.

We examined whether Drosha cleavage of the *Nfib* 3′ UTR HP is dependent upon binding to SAFB. Therefore, we generated a SAFB mutant lacking 156 amino acids containing the RE-rich region (aa552 to aa708; SAFB-ΔRE) (*Figure 5E*). The RE-rich region of SAFB2 has been shown to bind Drosha (*Hutter et al., 2020*). To assess Drosha binding to the SAFB-ΔRE protein, we tagged the mutant and WT SAFB variants with HA tags to distinguish them from endogenous protein, expressed these in neuroblastoma cells (N2A), and performed a Drosha co-IP followed by anti-HA immunoblot. We quantified the amount of SAFB WT or SAFB-ΔRE mutant proteins that co-precipitated with Drosha, and found that SAFB-ΔRE showed a 5.7-fold reduction in binding to Drosha compared to SAFB WT protein (*Figure 5F and G* and *Figure 5—figure supplement 1G and H*). We addressed the functional consequences of deleting the RE domain of SAFB on its regulation of the *Nfib* 3′ UTR by expressing the SAFB-ΔRE mutant in Tet-on 3′ UTR HP and Tet-on ctrl DG NSCs. Strikingly, and unlike overexpression of the WT HA-tagged SAFB protein, overexpression of the SAFB-ΔRE mutant did not reduce the expression of GFP in Tet-on 3′ UTR HP compared to Tet-on ctrl DG NSCs (*Figure 5H*), even though the SAFB-ΔRE protein was expressed at similar levels to the WT SAFB (*Figure 5—figure supplement 1I*). These data indicated that the SAFB RE domain is involved in SAFB binding to Drosha and enhances Drosha activity toward the *Nfib* 3′ UTR HP.

## SAFB represses *Nfib* expression in DG NSCs

SAFB reduced the expression of the *Nfib* 3′ UTR HP reporter in a Drosha-dependent manner. Therefore, we mapped SAFB RNA-binding motifs on the *Nfib* mRNA and found several potential binding sites in the 3′ UTR HP and flanking sequence (*Figure 6A*; *Rivers et al., 2015*; *Van Nostrand et al., 2020*). The presence of SAFB-binding sites on the *Nfib* mRNA in the 3′ UTR HP and flanking sequence is supported by the finding that *Nfib* mRNA is a SAFB target in HepG cells. The SAFB interaction domain on the *Nfib* mRNA was mapped to the 3′ UTR HP (*Figure 6A* and *Figure 6—figure supplement 1A*; *Van Nostrand et al., 2020*). We evaluated SAFB binding to the endogenous *Nfib* mRNA by crosslinking and immunoprecipitation (CLIP) with anti-SAFB antibodies from DG NSCs followed by RT-qPCR analysis (*Figure 6B*). Both *Nfib* mRNA and a known SAFB target, *Hnrnpu* mRNA, were bound and precipitated together with SAFB (*Figure 6C*). Therefore, we evaluated the effects of SAFB overexpression on endogenous *Nfib* expression in DG NSCs and found a significant reduction in *Nfib* mRNA levels (*Figure 6D*; *Knuckles et al., 2012*; *Rolando et al., 2016*). Thus, SAFB directly binds *Nfib* mRNA and reduces its levels in DG NSCs.

As SAFB promotes Drosha cleavage of *Nfib* transcripts, and Drosha activity controls DG NSC fate in vivo and in vitro (*Rolando et al., 2016*), we attempted to address the role of SAFB in DG NSCs by performing an esiRNA-mediated knockdown (KD) in vitro. Safb KD led to a rapid increase in activated CASP3 and cell death of DG NSCs within 48 hr, preventing further fate analysis (*Figure 6—figure supplement 1B*). Therefore, we turned to N2A cells that express Drosha, SAFB, and *Nfib* mRNA. esiRNA-mediated *Safb* KD caused an increase in *Nfib* mRNA levels in N2A cells supporting the

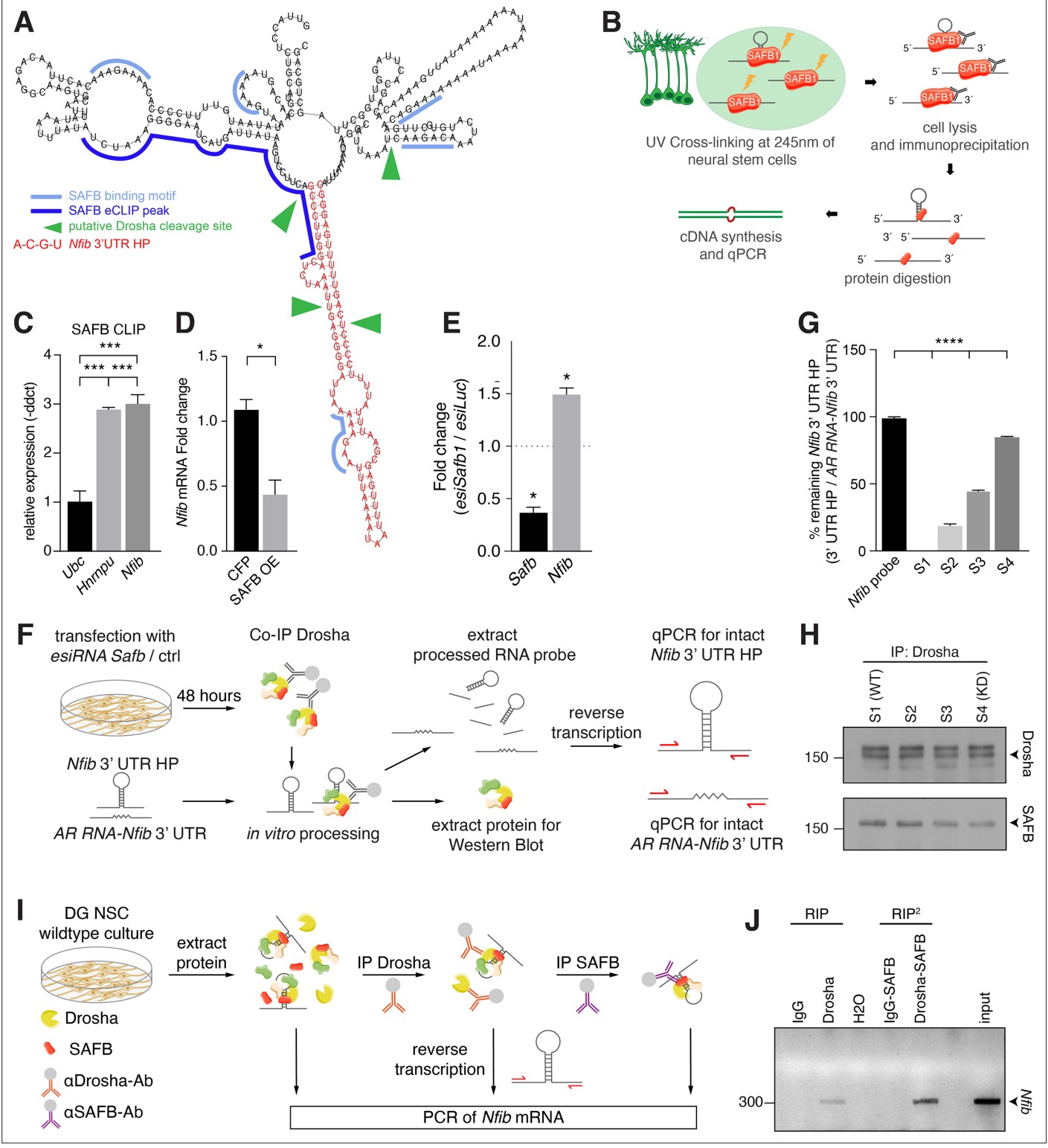

**Figure 6.** SAFB binds and regulates endogenous *Nfib* expression. (**A**) In silico motif analysis of SAFB-binding sites (light and dark blue) and the predicted secondary structure of the *Nfib* 3′ untranslated region (UTR) hairpin (HP) (red). SAFB crosslinking and immunoprecipitation (CLIP) peak mapped on the *Nfib* mRNA in HepG cells by ***Van Nostrand et al., 2020***. Putative Drosha cleavage sites (green arrowheads) mapped in ***Rolando et al., 2016***. (**B**) Scheme of the experimental setup for SAFB CLIP from dentate gyrus (DG) neural stem cells (NSCs) and detection of bound RNA transcripts by quantitative reverse transcriptase PCR (RT-qPCR). (**C**) RT-qPCR analysis of SAFB CLIP from DG NSCs. Relative levels of *Ubc*, *Hnrnpu*, and *Nfib* pull-down (-ddct values) calculated over input and minus antibody (-Ab) control. Negative control: *Ubc*, positive control: *Hnrnpu*. n=3, one-way ANOVA with

*Figure 6 continued on next page*

*Figure 6 continued*

Holm-Sidak's test: ***p<0.001. Error bars SEM. (**D**) RT-qPCR analysis of *Nfib* mRNA levels in DG NSCs transfected with *Cfp* or *Safb* overexpression (OE) vectors displayed as fold change *Nfib* mRNA over untransfected cells. n=4, two-tailed Mann-Whitney test: *p<0.05. Error bars SEM. (**E**) Quantification of relative *Safb* and *Nfib* mRNA expression levels in *Safb* knockdown (*esiSafb*) N2A cells after 48 hr compared to control (*esiLuc*) transfected cells. Two-tailed Student's t-test: *p<0.05. Error bars SEM. (**F**) Scheme of the experimental setup for Drosha complex immunoprecipitation from N2A cells with/without *Safb* knockdown (*esiSafb*) and in vitro processing of *Nfib* 3′ UTR HP and AR RNA-*Nfib* 3′ UTR (control) RNA probes. Drosha complexes were immunoprecipitated with anti-Drosha antibodies under native conditions from *Safb* esiRNA knockdown (esiRNA *Safb*) N2A and untransfected (control) N2A cells. Drosha complexes from control and *Safb* knockdown cells were mixed (ratios 100:0, 66:33, 33:66, and 0:100) and incubated with 500 ng *Nfib* 3′ UTR HP RNA or AR RNA-*Nfib* 3′ UTR RNA probe for 30 min. Proteins and RNA were extracted and analyzed by immunoblot and RT-qPCR for intact *Nfib* 3′ UTR HP RNA and AR RNA-*Nfib* 3′ UTR RNA. Both the *Nfib* 3′ UTR HP RNA and AR RNA-*Nfib* 3′ UTR RNA probe share the same primer binding sequences. Although the in vitro processing experiment was performed in the presence of RNase inhibitor, the AR RNA-*Nfib* 3′ UTR RNA probe served as a control for non-specific RNase activity. (**G**) RT-qPCR analysis of intact *Nfib* 3′ UTR HP RNA probe after incubation with Drosha complexes (S1–S4). Unprocessed probe *Nfib* 3′ UTR HP (*Nfib* probe) was used to calculate the percent remaining intact probe in samples S1–S4. The levels of intact *Nfib* 3′ UTR HP probe were normalized to the levels of AR RNA-*Nfib* 3′ UTR hybrid probe (AR RNA-*Nfib* 3′ UTR) incubated with the same lysates and compared to input unprocessed *Nfib* probe levels (see ***Figure 6—figure supplement 1F and G***). n=3, one-way ANOVA with Holm-Sidak's test: ****p<0.0001. Error bars SEM. (**H**) Immunoblot analysis of Drosha and SAFB in the Drosha IP (immunoprecipitation) complexes in samples S1–S4 of the in vitro Nfib 3′ UTR HP and AR RNA-*Nfib* 3′ UTR probe processing experiments. Sample S1 has endogenous levels of SAFB, S4 is a SAFB KD (knockdown) sample. S2 and S3 are mixes of S1 and S4 in ratios 66:33 and 33:66, respectively. Drosha levels were constant in S1–S4. (**I**) Experimental scheme of tandem RNA-immunoprecipitation (RIP[2]), precipitating *Nfib* transcript with anti-Drosha and subsequentially with anti-SAFB antibodies in two consecutive RNA-immunoprecipitations (RIP). Samples for RT-PCR were analyzed after cell lysis (input), the first RIP (RIP) and reRIP (RIP[2]). (**J**) Gel electrophoresis of non-quantitative PCR samples from input, RIP, and RIP[2] with specific amplicons for endogenous *Nfib* transcript (314 bp) in the respective sample lanes as opposed to IgG controls.

The online version of this article includes the following source data and figure supplement(s) for figure 6:

**Source data 1.** Original data for ***Figure 6J***.

**Source data 2.** Original data for ***Figure 6J***.

**Figure supplement 1.** SAFB contributes to regulation of *Nfib* mRNA level.

**Figure supplement 1—source data 1.** Original data for ***Figure 6—figure supplement 1C, F and G***.

**Figure supplement 1—source data 2.** Original data for ***Figure 6—figure supplement 1C, F and G***.

importance of SAFB for efficient *Nfib* mRNA processing (***Figure 6E*** and ***Figure 6—figure supplement 1C***).

We assessed whether SAFB directly affects *Nfib* 3′ UTR HP cleavage by performing a modified in vitro processing assay (***Rolando et al., 2016***). An in vitro transcribed *Nfib* 3′ UTR HP RNA probe was incubated with Drosha complexes immunoprecipitated from N2A cells and the presence of intact probe was quantified by RT-qPCR (***Figure 6F***). Additionally, we employed the control AR RNA-*Nfib* 3′ UTR probe in these assays. The control AR RNA-*Nfib* 3′ UTR probe includes the regions surrounding the *Nfib* 3′ UTR HP, but the HP-forming sequence is substituted with the AR sequence that is not susceptible to cleavage by Drosha. The *Nfib* 3′ UTR HP and AR RNA-*Nfib* 3′ UTR probes share the same polymerase chain reaction (PCR) primer sequences allowing the same PCR conditions to be used to quantify both probes. The AR RNA-*Nfib* 3′ UTR probe also allowed us to monitor any non-specific RNase activity in the assay. The *Nfib* 3′ UTR HP probe was completely processed following incubation with Drosha complexes and the full-length intact probe was not detectable (S1 [WT] in ***Figure 6G*** and ***Figure 6—figure supplement 1D***). However, the AR RNA-*Nfib* 3′ UTR control probe remained intact throughout the incubation (S1 [WT] in ***Figure 6—figure supplement 1E***).

To address whether the levels of SAFB affect Drosha processing of the *Nfib* 3′ UTR HP, we immunoprecipitated Drosha complexes from *Safb* KD N2A cells and mixed these in ratios of 33:66, 66:33, and 100:0 (S2, S3, and S4 [KD]) with Drosha complexes immunoprecipitated from N2A cells expressing WT levels of SAFB. We quantified the levels of SAFB and Drosha in each reaction by immunoblot which revealed the predicted decrease in SAFB levels from reactions S1 (WT) to S4 (KD), while Drosha levels remained equal (***Figure 6H*** and ***Figure 6—figure supplement 1F and G***). Drosha complexes from *Safb* KD N2A cells did not cleave the *Nfib* 3′ UTR HP probe, which remained intact over the incubation period (S4 [KD] in ***Figure 6G***). Furthermore, reducing the levels of SAFB in the in vitro processing reaction perturbed the processing of the *Nfib* 3′ UTR HP probe without affecting AR RNA-*Nfib* 3′ UTR probe integrity (***Figure 6G*** and ***Figure 6—figure supplement 1D and E***). Thus, *Nfib* 3′ UTR HP

processing by Drosha shows a direct relationship to the levels of SAFB in the complex (*Figure 6G and H*).

## SAFB and Drosha bind to the same *Nfib* transcript in DG NSCs

In order to investigate whether Drosha and SAFB interact directly on the same native *Nfib* transcript, we developed a tandem RNA-immunoprecipitation (RIP[2]) assay based on chromatin re-immunoprecipitation (reChIP) (*Figure 6I*). We adapted the reChIP protocol for RIP[2]. We performed a first RNA-immunoprecipitation with anti-Drosha or control IgG antibodies (RIP assay). The precipitates were eluted and subjected to a subsequent RNA-immunoprecipitation using anti-SAFB or control IgG antibodies (RIP[2]). The final precipitate was eluted and *Nfib* 3' UTR transcripts were detected by RT-PCR. *Nfib* 3' UTR amplicons were detected in the RIP and RIP[2] samples precipitated with anti-Drosha and anti-SAFB antibodies but not in the control IgG precipitates (*Figure 6J*). These results support simultaneous interaction of Drosha and SAFB on the same native *Nfib* mRNA in DG NSCs.

## SAFB regulates oligodendrocyte differentiation from NSCs

DG NSCs predominately generate granule neurons and astrocytes but not oligodendrocytes, and their fate restriction is controlled in part by Drosha through post-transcriptional repression of NFIB expression (*Rolando et al., 2016*). *Hes5* is a Notch signaling target and is expressed by NSCs in the DG and Hes5::CreER[T2] activity can be used to lineage trace neurogenic and gliogenic cells in the adult mouse (*Lugert et al., 2010*; *Lugert et al., 2012*). We labeled NSCs in the adult mouse brain by treating Hes5::CreER[T2] mice carrying a Rosa26-CAG::Egfp Cre-reporter for 5 days with Tamoxifen.

To assess SAFB expression in the adult mouse brain, we immunostained brain sections of Tamoxifen-induced Hes5::CreER[T2] Rosa26-CAG::Egfp mice with anti-SAFB antibodies (*Lugert et al., 2012*). SAFB is expressed by many cells in the adult DG and V-SVZ including GFAP[+], *Hes5*[+] NSCs (*Figure 7A*). SAFB levels were significantly higher in neurons than in astrocytes (GFAP[+]) and oligodendrocytes (SOX10[+]) in the DG and striatum adjacent to the V-SVZ of the lateral ventricle wall (*Figure 7A–C*). As NFIB is associated with glial differentiation, we hypothesized that *Nfib* mRNA processing should be lower in astrocytes than in multipotent NSCs. We used the Tet-on 3' UTR HP reporter system and compared GFP levels in undifferentiated NSCs and NSCs differentiated into astrocytes following treatment with calf serum (*Figure 7D*). GFP levels expressed from the Tet-on 3' UTR HP reporter were increased in astrocytes compared to undifferentiated NSCs relative to the Tet-on ctrl reporter expression (*Figure 7D*).

In contrast to DG NSCs, V-SVZ NSCs generate oligodendrocytes in vitro and DG NSCs express higher levels of SAFB compared to V-SVZ NSCs in vitro (*Figure 7A and E* and *Figure 7—figure supplement 1A and F*). We assessed the role of SAFB in NSC fate choice decisions by performing gain-of-function experiments in adult V-SVZ NSCs. Overexpression of SAFB (CAG::Safb-IRES-Cfp) in V-SVZ NSCs reduced oligodendrocyte (SOX10[+]CFP[+]) production compared to cells expressing the control vector (CAG::IRES-Cfp) (SAFB overexpression: 19.9 ± 1.6% versus CFP ctrl: 33.2 ± 1.2%: p<0.01) and non-transfected CFP[-] cells (*Figure 7F–H* and *Figure 7—figure supplement 1B*). Quantification of CFP[+]GFAP[+] cells revealed no differences in the astrocytic differentiation of SAFB overexpressing cells compared to CAG::IRES-Cfp expressing and non-transfected CFP[-] cells (*Figure 7—figure supplement 1C–E*). Therefore, SAFB overexpression repressed oligodendrocytic differentiation of multipotent V-SVZ NSCs in a cell autonomous fashion supporting that SAFB levels regulate NSC fate choices.

## Conditional ablation of SAFB increases oligodendrocyte generation in the hippocampus of adult mice

We showed by gain of function that SAFB regulates oligodendrocyte production by adult NSCs in vitro. In order to address the role of SAFB in vivo, we generated a floxed *Safb* allele (*Safb*[fl/fl]) and conditionally knocked out the *Safb* gene from adult mice using Hes5::CreER[T2] and followed that fate of *Safb* cKO DG NSCs and their progeny by tracing Rosa26-CAG::Egfp Cre-reporter positive cells (*Figure 8A*). 21 days after *Safb* cKO, GFP[+]SOX10[+] and GFP[+]OLIG2[+] cells were significantly increased in the DG of *Safb* cKO mice (*Safb*[fl/fl] Hes5::CreER[T2] Rosa26-CAG::Egfp; 2.1-fold increase [p=0.006]) compared to control animals (*Safb*[+/+] Hes5::CreER[T2] Rosa26-CAG::Egfp) (*Figure 8A–C*). The increase of GFP[+] oligodendrocyte marker positive cells was also evident in DG of heterozygous mice (*Safb*[fl/+] Hes5::CreER[T2] Rosa26-CAG::Egfp; 1.6-fold increase [p=0.009]) compared to control animals suggesting a dosage

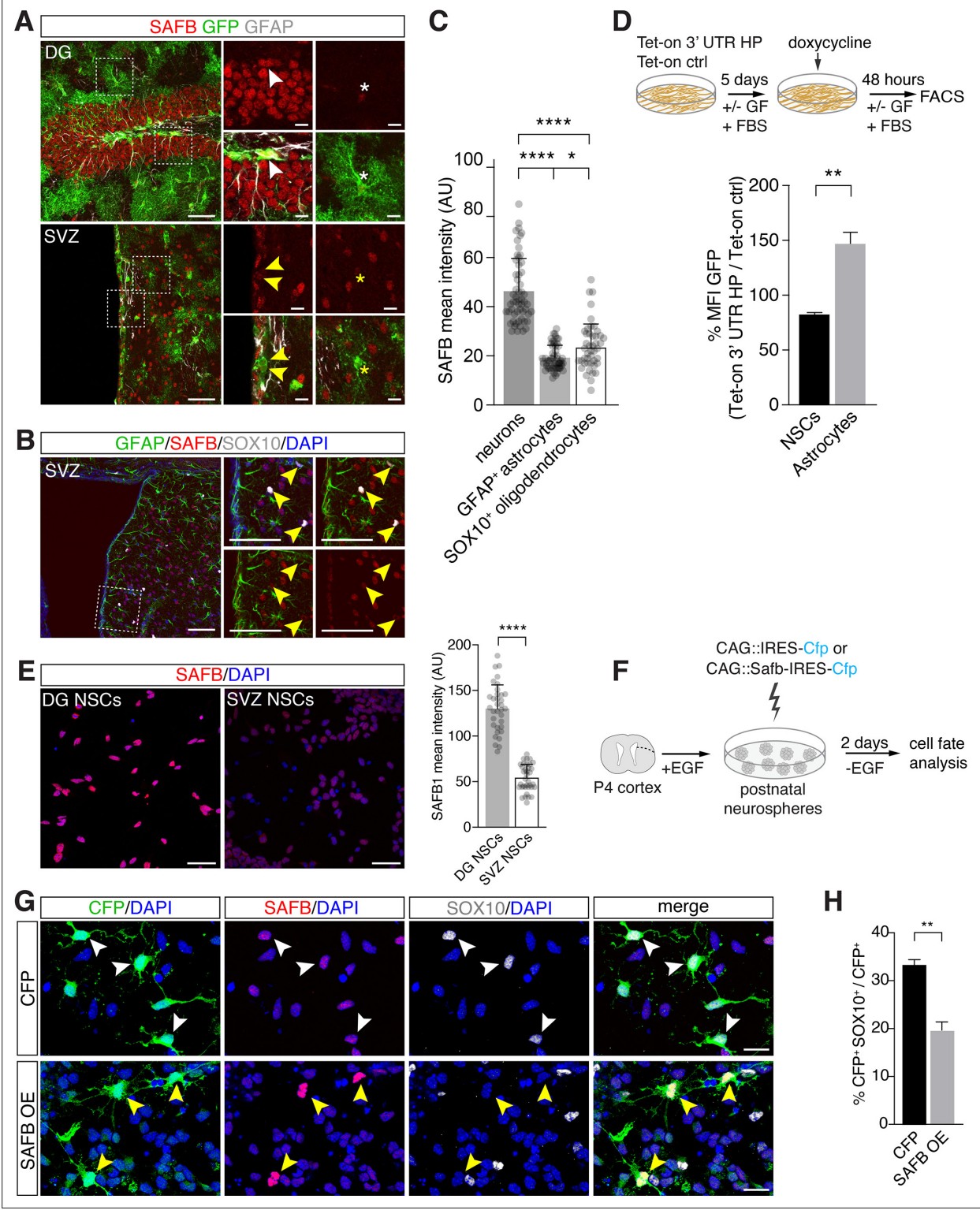

**Figure 7.** SAFB overexpression regulates oligodendrogenesis. (**A**) SAFB expression in adult HES5+ dentate gyrus (DG) neural stem cell (NSC) marked with Hes5::CreER[T2] and Rosa26-CAG::Egfp Cre-reporter (upper panels white arrowheads) and ventricular-subventricular zone (V-SVZ) NSCs genetically labeled with Hes5::CreER[T2] and Rosa26-CAG::Egfp Cre-reporter (lower panels yellow arrowheads) in vivo. NSCs were labeled in vivo by Tamoxifen treatment of Hes5::CreER[T2] Rosa26-CAG::Egfp mice (see Materials and methods). Regions of high-magnification images are indicated. DG NSCs (GFP+; white arrowheads upper panels) and DG neurons (GFP- cells) express higher levels of SAFB than astrocytes (white *). V-SVZ NSCs (GFP+; yellow arrowheads lower panels) and astrocytes (yellow *) express lower levels of SAFB than striatal neurons (GFP- cells). Scale bars: low-magnification images

*Figure 7 continued on next page*

*Figure 7 continued*

100 μm, high-magnification images 20 μm. (**B**) V-SVZ of adult wild type mouse with astrocytic GFAP, oligodendrocytic SOX10, nuclear DAPI and SAFB labeling. High-magnification images correspond to region indicated. SOX10$^+$ SVZ cells show low levels of SAFB (yellow arrowheads). Scale bars 100 μm. (**C**) Quantification of SAFB expression by neurons, astrocytes, and oligodendrocytes in vivo based on mean intensity levels (arbitrary units). One-way ANOVA with Holm-Sidak's test: ****$p<0.0001$, ns - not significant. Error bars SEM. (**D**) Scheme of the experimental setup for quantification of EGFPd2 (GFP) protein fluorescence comparing Tet-on 3' untranslated region (UTR) hairpin (HP) over Tet-on ctrl DG NSCs by FACS in undifferentiated NSC state and after growth factor (GF) removal and addition of fetal bovine serum (FBS) for 5 days to induce differentiation to astrocytes followed by a 48 hr doxycycline induction. Quantification of the median EGFPd2 (GFP) intensity of Tet-on 3' UTR HP relative to Tet-on ctrl in NSCs and astrocytes. n=3, two-tailed Student's t-test: **$p<0.01$. Error bars SEM. (**E**) Images and quantification of SAFB protein expression by adult DG NSCs and V-SVZ NSCs in vitro. Scale bar 20 μm. Kolmogorov-Smirnov test: ****$p<0.0001$. Error bars SEM. Scale bars 50 μm. (**F**) Scheme for the experimental setup for SAFB (CAG::Safb-IRES-Cfp) and control CFP (CAG::IRES-Cfp) overexpression in postnatal V-SVZ NSCs (postnatal day 4: P4) grown as neurospheres in the presence of EGF and cell fate analysis 2 days after EGF withdrawal (w/o EGF) to induce differentiation. (**G**) Analysis of the effects of SAFB overexpression (OE) in V-SVZ neurospheres and effects on oligodendrocyte (SOX10$^+$) differentiation compared to CFP (control) expression. Scale bars 20 μm. (**H**) Quantification of the percentage of transfected cells (CFP$^+$) expressing SOX10 (oligodendrocytes) after SAFB overexpression (OE) (n=4) or CFP alone (n=3). Two-tailed t-test with Welch's correction: **$p<0.01$. Error bars SEM.

The online version of this article includes the following source data and figure supplement(s) for figure 7:

**Figure supplement 1.** SAFB overexpression (OE) does not increase astrocyte differentiation.

**Figure supplement 1—source data 1.** Original data for *Figure 7—figure supplement 1A*.

**Figure supplement 1—source data 2.** Original data for *Figure 7—figure supplement 1A*.

sensitivity of DG NSCs to SAFB levels (*Figure 8B and C*). As expected, GFP$^+$ cells in the SGZ of *Safb* cKO mice had undetectable levels of SAFB protein compared to those in control mice (*Figure 8E*).

As oligodendrocyte progenitors can migrate extensively, we analyzed the distribution of GFP$^+$SOX-10$^+$OLIG2$^+$ oligodendrocytes in the DG of *Safb* cKO compared to control mice. GFP$^+$SOX10$^+$OLIG2$^+$ cells were significantly enriched in the SGZ *Safb* cKO mice compared to controls (p=0.018). However, this enrichment in adult NSC-derived oligodendrocytes was even stronger in the hilus (p=0.005) but was not changed in the granule cell layer (GCL; p=0.337) and molecular layer (ML; p=0.282) (*Figure 8B and D*). This suggested that deletion of *Safb* increased oligodendrocyte production in the adult DG and that these SOX10$^+$OLIG2$^+$ cells migrated preferentially into the hilus region in the DG. We addressed the effects of *Safb* cKO on NFIB expression in the DG. We immunostained the DG of *Safb* cKO and control mice and quantified the relative expression levels of NFIB expressed by progenitors in the SGZ (*Figure 8E and F*). NFIB levels in GFP$^+$ SGZ cells of *Safb* cKO mice were increased 1.8-fold compared to control mice (*Figure 8E and F*). Therefore, loss of SAFB correlated with an increase in NFIB levels and an increase in the production of oligodendrocytes.

## Discussion

The control of stem cell fate is critical for tissue development and homeostasis. The ribonuclease Drosha is the catalytic component of the miRNA Microprocessor and forms a dimer with the RBP DGCR8 (*Han et al., 2004*; *Nguyen et al., 2015*). However, Drosha can directly cleave mRNAs, including those encoding transcription factors critical for NSC maintenance and differentiation (*Kim et al., 2017*; *Rolando and Taylor, 2017*; *Chong et al., 2010*; *Knuckles et al., 2012*; *Rolando et al., 2016*; *Han et al., 2009*). However, it is unclear how the specificity of Drosha activity toward its target mRNAs is determined and regulated in a cell-specific fashion.

Drosha-associated proteins form megadalton complexes containing more than 20 different proteins (*Nguyen et al., 2015*; *Macias et al., 2015*). The functions of the large Drosha complexes and the associated proteins are mostly unknown but it is likely that they contribute to regulating Drosha activity and cell type-specific target specificity. In the adult mouse DG, Drosha modulates NFIB expression by binding to and cleaving its mRNA at an evolutionarily conserved 3' UTR HP. The suppression of NFIB expression restricts adult DG NSC fate and prevents oligodendrocyte formation (*Rolando et al., 2016*).

As Drosha is ubiquitously expressed, we hypothesized that the catalytic activity on the NFIB mRNA in DG NSCs is controlled by partner proteins which provide sequence specificity and direct cleavage. We undertook a proteomic approach and identified proteins that interact with endogenous Drosha in DG NSCs. Most of the 165 Drosha-interacting proteins we identified are RBPs, consistent with their

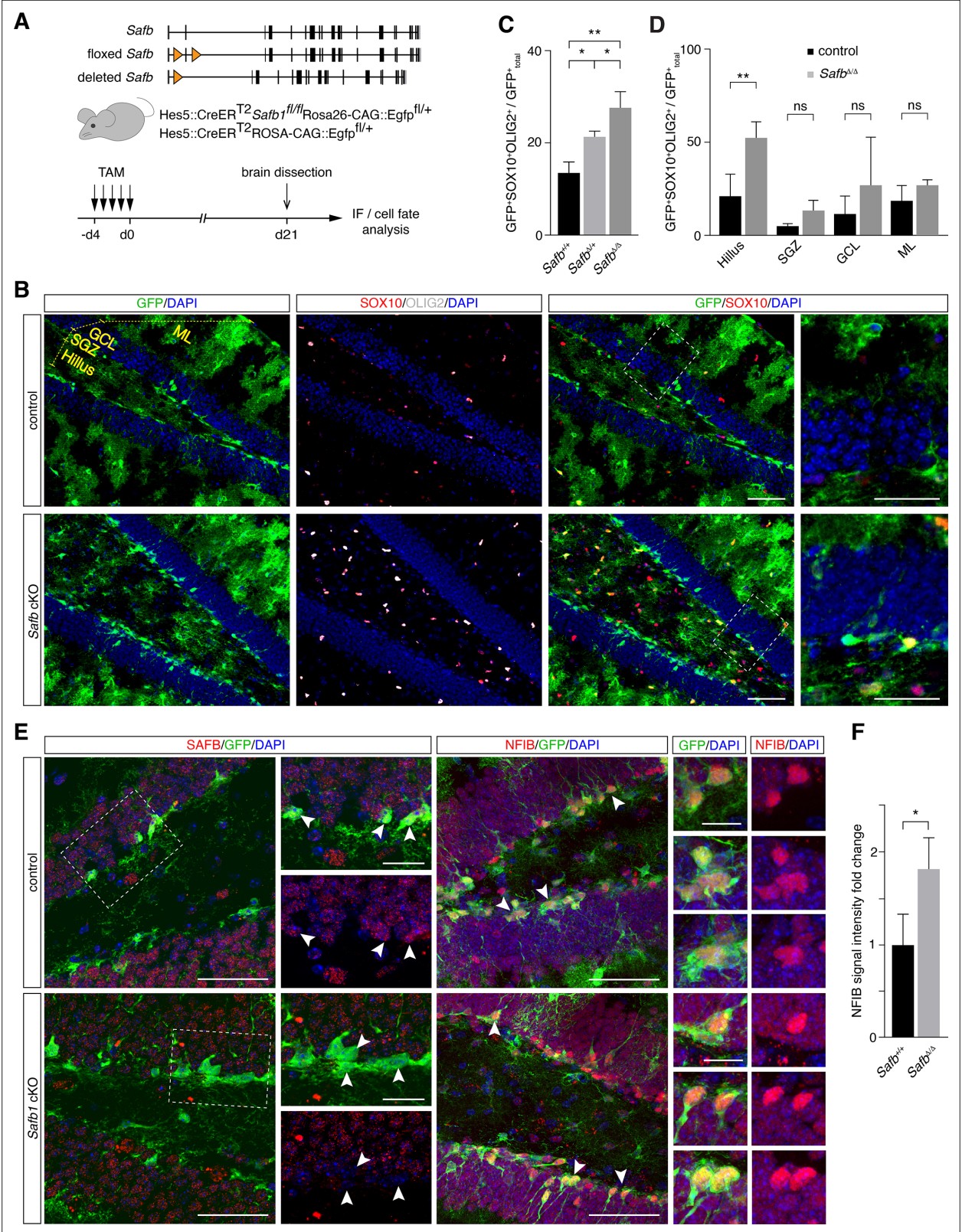

**Figure 8.** *Safb* cKO in neural stem cells (NSCs) increases oligodendrogenic fate in the adult hippocampus. (**A**) Exon II floxed *Safb* (*Safb^fl/fl*) mice were bred with Hes5::CreER^T2 Rosa26-CAG::Egfp Cre-reporter mice to generate *Safb^fl/fl* Hes5::CreER^T2 Rosa26-CAG::Egfp and *Safb^+/+* Hes5::CreER^T2 Rosa26-CAG::Egfp control animals. Cre-recombinase deletes the exon II of the *Safb* gene resulting in a null allele and expression of GFP from the Rosa26-CAG::Egfp allele to trace the fate of the NSCs and their progeny. Eight-week-old adult mice were treated with Tamoxifen (TAM) by intraperitoneal

*Figure 8 continued on next page*

*Figure 8 continued*

injection for 5 consecutive days and the animals sacrificed 21 days later for analysis by immunofluorescence (IF). (**B**) Representative immunofluorescent images of the dentate gyrus (DG) stained with antibodies against GFP, SOX10, and OLIG2 and counterstained with DAPI. Scale bar low-magnification images: 50 μm, high-magnification images: 20 μm. High-magnification images correspond to regions indicated by boxes in the lower magnification images. Subdivision of DG compartments indicated and labeled in yellow. (**C**) Quantification of GFP+SOX10+OLIG2+ triple-positive cells over total GFP+ cells (arbitrary units). Statistical comparison of GFP+SOX10+OLIG2+ oligodendrocytes over total GFP+ cells in control (*Safb*+/+), heterozygous (*Safb*Δ/+), and homozygous (*Safb*Δ/Δ) mice. Δ - Cre-deleted allele. ANOVA with Holm-Sidak's test: *p<0.05, **p<0.01. Error bars SEM. (**D**) Subdivision and quantification in distinct compartments of the DG comparing GFP+SOX10+OLIG2+ oligodendrocytes over total GFP+ in control and *Safb* cKO (homozygous *Safb*Δ/Δ) mice. Individual two-tailed t-test with Welch's correction: **p<0.01, ns: not significant. Error bars SEM. *Safb* cKO: conditional knockout. (**E**) Immunofluorescent images of the DG of control and *Safb* cKO mice stained with antibodies against SAFB, GFP, and NFIB and counterstained with DAPI. GFP-expressing cells are Hes5::CreER^T2-expressing NSC (white arrowheads) and their progeny. Scale bar low-magnification images: 50 μm, high-magnification images: 20 μm. High-magnification images correspond to regions indicated by boxes in the lower magnification images. (**F**) Quantification of the relative NFIB signal intensity in GFP+ cells in the DG of control (*Safb*+/+) and *Safb* cKO (*Safb*Δ/Δ) animals (arbitrary units). Two-tailed t-test with Welch's correction: *p<0.05. Error bars SEM.

potential to modulate Drosha activity on its RNA substrates (*Gerstberger et al., 2014*; *Huang et al., 2018*). However, some of these proteins have not been shown to bind RNA and may be components of large macromolecular complexes that could play diverse regulatory functions on Drosha activity. 15% of the Drosha-binding proteins in DG NSCs have also been shown to bind Drosha in HEK293T cells indicating a degree of molecular conservation across cell types but also suggesting cell-type specificity of the Drosha interactome (*Rouillard et al., 2016*; *Macias et al., 2015*). Although the roles of most of these Drosha-interacting proteins in NSCs are not known, our data provide insights into the complexity and potential novel functions of Drosha signaling.

NFIB is a critical transcription factor in glial cell differentiation. In the adult DG, NSCs retain tri-lineage potency but differentiation to oligodendrocytes is blocked by suppression of NFIB protein expression (*Rolando et al., 2016*). Drosha destabilizes *Nfib* mRNAs in DG NSCs and, as a consequence, promotes neurogenesis. The *Nfib* mRNA contains two evolutionary conserved HPs located in the 5′ and 3′ UTR, both of which are bound by Drosha but only the 3′ UTR HP is cleaved by Drosha (*Rolando et al., 2016*). We found that DGCR8, the partner of Drosha in early miRNA biogenesis, does not interact with the *Nfib* UTR HPs and DGCR8 overexpression did not affect *Nfib* mRNA cleavage. Therefore, DGCR8 and the core Microprocessor are unlikely to contribute directly to Drosha-mediated regulation of *Nfib* expression by adult NSCs. This implied the requirement for other factors in cell type-specific regulation of *Nfib* mRNA stability.

Therefore, we used the *Nfib* mRNA 5′ UTR and 3′ UTR HPs as baits to capture interacting DG NSC proteins. Comparison of the MS² data from these pull-down experiments revealed major differences in the proteins associated with the 5′ UTR and 3′ UTR HPs suggesting different functions of these sequences in the control of NFIB expression. The *Nfib* 5′ UTR HP interacted with ribosomal proteins including RPL7, RPL15, RPL23a, RPL27, RPL31, RPS7, RPS15a, and RPS25 suggesting a role for the 5′ UTR HP in NFIB translation. This is supported by the GO analysis of the 5′ UTR HP-associated proteins suggesting translational regulation. Whether Drosha plays a role in NFIB translation through the 5′ UTR HP region of its mRNA remains unclear.

We considered that proteins that preferentially associate with the 3′ UTR HP of *Nfib* mRNA potentially regulate the cleavage activity of Drosha at this site. We found that all three SAFB protein family members, SAFB-like transcription modulator (SLTM), SAFB, and SAFB2 bind to the *Nfib* mRNA. STLM, SAFB, and SAFB2 were also pulled down from NSC lysates together with Drosha. As SAFB proteins form homo- and heterodimers, it is possible that they can form distinct multimeric complexes with Drosha and convey different functions. Interestingly, although broadly expressed, SAFB proteins are not redundant as SAFB-deficient mice die during embryonic development or perinatally whereas SAFB2-null mice are viable (*Ivanova et al., 2005*; *Jiang et al., 2015*). In contrast to SAFB, SLTM and SAFB2 do not bind selectively to the *Nfib* 3′ UTR HP. SAFB2 has been shown to bind to Drosha in human cells where it assists in the processing of the non-optimal miRNA15a/16-1 pri-miRNA cluster, although this function does not require the SAFB2 RNA-binding domain (*Hutter et al., 2020*). Hence, it will be important to also investigate the roles of SLTM and SAFB2 in DG NSC fate regulation.

SAFB has not been shown previously to bind to Drosha and is a novel partner of Drosha in adult DG NSCs where it regulates oligodendrogenesis both in vitro and in vivo by preventing NFIB expression. We showed that SAFB levels are important in controlling Drosha cleavage of *Nfib* mRNA in a

dose-dependent fashion. This is in line with the SAFB expression by adult DG NSCs, where oligoden-drogenesis does not occur under physiological conditions, and lower expression by oligodendrogenic NSCs of the postnatal V-SVZ (*Rivers et al., 2015*).

SAFB binds AT-rich scaffold-matrix attachment regions in DNA and GAA/AAG/AGA triplicates in a GAAGA consensus motif on RNA to regulate RNA-dependent chromatin organization and mRNA processing (*Rivers et al., 2015*; *Van Nostrand et al., 2020*; *Stoilov et al., 2004*; *Huo et al., 2020*; *Renz and Fackelmayer, 1996*). SAFB binding to the *Nfib* 3′ UTR HP but not to the *Nfib* 5′ UTR HP indicates sequence specificity, and we mapped putative SAFB-binding sites around the 3′ UTR HP which is supported by CLIP analysis of SAFB binding to *Nfib* mRNA in HepG cells (*Van Nostrand et al., 2020*). Using RIP[2] we show that Drosha and SAFB interact with the same endogenous *Nfib* mRNA transcript. In addition, a mutant form of SAFB lacking the RE protein interaction domain which does not bind Drosha did not promote cleavage of *Nfib* mRNA transcripts. Together these data show that Drosha cleavage of *Nfib* mRNA transcripts in DG NSCs requires SAFB binding and that both proteins associate simultaneously with the same transcript.

Here, we focused on SAFB, but found many novel Drosha-binding proteins that associated with the *Nfib* UTR HPs. Interestingly, SAM68, a member of the STAR family of RBPs, also affected *Nfib* HP processing. SAM68 has been shown to control RNA splicing also during neurogenesis (*Chawla et al., 2009*). SAFB interacts with SAM68 and they co-distribute in the nucleus in response to stress (*Sergeant et al., 2007*). SAM68 has been shown to interact with the Microprocessor in different systems (*Messina et al., 2012*; *Sellier et al., 2013*). It is currently unclear whether SAM68 controls Drosha canonical or non-canonical functions potentially together with SAFB. We found that in contrast to SAFB, FUS enhanced the levels of GFP expression from the *Nfib* 3′ UTR HP reporter. Interestingly, FUS also interacts with SAFB on DNA (*Yamaguchi and Takanashi, 2016*). This raises the intriguing question of whether FUS and SAFB cooperate to regulate *Nfib* mRNA or work in independent protein complexes.

In summary, several mRNAs are now known to be targets of Drosha and most of them contain evolu-tionarily conserved HP structures (*Rolando and Taylor, 2017*; *Chong et al., 2010*; *Knuckles et al., 2012*; *Rolando et al., 2016*; *Johanson et al., 2013*). More than 2000 human mRNAs are predicted to form secondary HPs that resemble pri-miRNAs, opening up a huge potential for non-canonical Drosha mRNA endonuclease function in different cells (*Johanson et al., 2013*; *Pedersen et al., 2006*). Given the wide range of Drosha-interacting RBPs, it is conceivable that Drosha targets many more mRNAs than is currently appreciated and thus could control many biological processes. We propose that Drosha-mediated RNA cleavage requires interactions with specific RBPs that direct target specificity and activity potentially in a contextual and cell type-specific fashion. A differential targeting of Drosha to cell type-specific substrates has major implications in the dynamic control of Drosha activity and functions. Therefore, our data with in vivo implications uncovered novel Drosha-interacting proteins and in the future, it will be important to investigate these Drosha/RBP complexes and their functional involvement in NSC fate regulation as well as in other cellular systems and contexts.

## Materials and methods
### Mouse strains and husbandry

The mouse alleles used in our experiments were: Rosa26-CAG::Egfp, Hes5::CreER[T2] (*Rolando et al., 2016*; *Lugert et al., 2010*), *Drosha^{fl/fl}* (*Chong et al., 2010*), and *Safb^{fl/fl}*. No gender differences were observed. Mice were randomly selected for the experiments based on birth date and genotype. According to Swiss Federal and Swiss Veterinary office regulations, all mice were bred and kept in a specific pathogen-free animal facility with 12 hr day-night cycle and free access to clean food and water. All mice were healthy and immunocompetent. All procedures were approved by the Basel Cantonal Veterinary Office under license number 2537 (Ethics commission Basel-Stadt, Basel Switzerland).

*Safb* floxed mice were generated by Cas9-assisted targeting of exon II of the *Safb* gene in C57BL6 mice. Guide RNAs (gRNAs) targeting sequences flanking *Safb* exon II were designed and selected based on their predicted target sites and lack of off-target sites. Ribonuclear particles containing Cas9 and the gRNAs (RNPs) were electroporated together with the template DNA containing the homolo-gous *Safb* exon II flanked by LoxP sequences into C57BL6 two-cell embryos. Injected embryos were implanted into surrogate females and resulting pups were genotyped to identify founder mice with

targeted *Safb* exon II alleles. Founders were mated with WT C57BL6 mice and animals with floxed *Safb* alleles interbred to generate lines. *Safb*$^{fl/fl}$ mice were bred with Hes5::CreER$^{T2}$ Rosa26-CAG::Egfp mice and back-bred to *Safb*$^{fl/fl}$ to generate *Safb*$^{fl/fl}$ Hes5::CreER$^{T2}$ Rosa26-CAG::Egfp mice. *Safb* cKO in adult NSCs was induced by intraperitoneal injection of 10-week-old *Safb*$^{fl/fl}$ Hes5::CreER$^{T2}$ Rosa26-CAG::Egfp, *Safb*$^{fl/wt}$ Hes5::CreER$^{T2}$ Rosa26-CAG::Egfp, and *Safb*$^{wt/wt}$ Hes5::CreER$^{T2}$ Rosa26-CAG::Egfp mice with Tamoxifen (2 µg/kg) once per day for 5 consecutive days. Mice were killed after a 21-day chase and perfused transcardially with 4% paraformaldehyde as described previously (*Rolando et al., 2016*). Animal strains and information are reported in the Key resources table.

## Hippocampal adult NSC cultures

DG NSCs were isolated, cultured, and differentiated as described previously (*Rolando et al., 2016*; *Babu et al., 2007*). 8- to 10-Week-old mice were sacrificed in a CO$_2$ chamber and decapitated. The brain was extracted, washed in ice-cold sterile L15 medium (Gibco), and live sectioned at 500 µm using a McIlwain's tissue chopper. Brain slices were collected in cold HBSS, 10 mM HEPES, and 100 IU/ml penicillin and 100 µg/ml streptomycin in a 6 cm culture dish. After careful micro-dissection of the DG and removing the molecular layer and ventricular zone contaminants using a dissection binocular microscope, the dissected DGs were collected in cold HBSS, 10 mM HEPES, and 100 IU/ml penicillin and 100 µg/ml streptomycin in a 15 ml Falcon tube. After tissue sedimentation, the supernatant was removed and replaced by 100 µl pre-warmed Papain mix. The tissue pieces were incubated at 37°C in a water bath for 15 min with gentle agitation every 5 min, followed by addition of 50 µl of pre-warmed Trypsin inhibitor and incubation for 10 min at 37°C. After adding 300 µl of DMEM/F12, the tissue was triturated with a 1 ml and 200 µl pipette tip. The sample was centrifuged at 80 × *g* to remove debris. The cell pellet was resuspended in DG NSC medium (DMEM/F12, Gibco, Invitrogen), 2% B27 (Gibco, Invitrogen), FGF 20 ng/ml (R&D Systems), EGF 20 ng/ml (R&D Systems), and plated in a 48-well dish (Costar) coated with 100 µg/ml poly-L-lysine (Sigma-Aldrich) and 1 µg/ml laminin (Sigma-Aldrich). Half of the cell medium was replaced every 2 days and cells were passaged every 6 days. Cells were passaged 6–20 times and characterized for marker expression and differentiation potential before use. Cell differentiation was induced by growth factor removal and continued culture for 6 days. Cell fixation for immunohistochemistry was performed for 10 min in 4% paraformaldehyde in 0.1 M phosphate buffer. Cell lines used in this study are reported in the Key resources table. N2A cells (CCL-131) were obtained from ATCC. The cells are routinely checked for negative mycoplasma contamination. Primary cells used in this study are reported in the Key resources table.

## Immunohistochemistry for brain tissues and NSC cultures

Mice were deeply anesthetized by injection of a ketamine/xylazine/acepromazine solution (150, 7.5, and 0.6 mg/kg body weight, respectively). Animals were perfused with ice-cold 0.9% saline followed by 4% paraformaldehyde in 0.1 M phosphate buffer. Brains were isolated and post-fixed overnight in 4% paraformaldehyde in 0.1 M phosphate buffer, and then cryoprotected with 30% sucrose in phosphate buffer at 4°C overnight. Brains were embedded and frozen in OCT (TissueTEK) and sectioned as 30 µm floating sections by cryostat (Leica). Free-floating coronal sections were stored at –20°C in antifreeze solution until use. DG NSCs fixation for immunohistochemistry was performed for 10 min in 4% paraformaldehyde in 0.1 M phosphate buffer.

Sections were incubated overnight at room temperature, with the primary antibody diluted in blocking solution of 1.5% normal donkey serum (Jackson ImmunoResearch), 0.5% Triton X-100 in phosphate-buffered saline. DG NSC cultures were incubated overnight at 4°C, with the primary antibody diluted in blocking solution of 1.5% normal donkey serum, 1% BSA, 0.2% Triton X-100 in phosphate-buffered saline.

Antibodies used: anti-activated cleavedCASP3 (Cell Signaling, rabbit, 1:500), anti-BLBP (Chemicon, rabbit, 1:400), anti-GFP (AbD Serotec, sheep, 1:250; Invitrogen, rabbit, 1:700; Aves Labs, chicken, 1:500), anti-MAP2 (Sigma-Aldrich, mouse, 1:200), anti-SAFB (Abcam, rabbit, 1:200), anti-SOX10 (Santa Cruz, goat, 1:500), anti-GFAP (Sigma-Aldrich, mouse, 1:200), anti-S100b (Sigma-Aldrich, mouse, 1:100), anti-NFIB (Invitrogen, rabbit, 1:1000), and anti-OLIG2 (Chemicon, rabbit, 1:300). Sections were washed in phosphate-buffered saline and incubated at room temperature for 2 hr with the corresponding secondary antibodies in blocking solution. DG NSCs were washed in 1% BSA

phosphate-buffered saline and incubated at room temperature for 35 min with the corresponding secondary antibodies in blocking solution.

Secondary antibodies and detection: Alexa488/Cy3/Alexa649-conjugated anti-chicken, mouse, goat, rabbit, and sheep immunoglobulin (1:600, Jackson ImmunoResearch). Sections were then washed and counterstained with DAPI (1 µg/ml). Stained sections were mounted on Superfrost glass slides (Thermo Fisher Scientific), embedded in mounting medium containing diazabicyclo-octane (DABCO; Sigma-Aldrich) as an anti-fading agent. Brain sections and DG NSCs were visualized using a Zeiss LSM510 confocal microscope, Leica SP5 confocal microscope, or Zeiss Apotome2 microscope. The antibody information is described in the Key resources table.

## Cell lysis for proteomics

DG NSCs were washed with PBS and incubated with lysis buffer (20 mM HEPES-KOH pH 7.9; 100 mM KCl; 0.2 mM EDTA; 0.2 mM PMSF, 1× cOmplete Protease Inhibitor (Roche); 5% glycerol; 0.5% Triton) for 10 min on ice. Cells were scraped off with cell scraper and lysate was transferred in Eppendorf tube, followed by sonication for five cycles (Bioruptor, 30 s on/30 s off, 320 W) at 4°C. Lysate was centrifuged at 13,000 × *g* for 10 min at 4°C. The supernatant was transferred in a new tube and BCA assay (Thermo Fisher Scientific, #23250) was performed according to the protocol of the manufacturer to determine protein concentration.

## Drosha co-IP assay

1 mg of total protein lysate was treated with or without RNase, incubated with antibody (anti-Drosha, Cell Signaling, rabbit, 1:50 or anti-SAFB, Abcam, rabbit, 1:50) on a rotating wheel overnight at 4°C (where 2.5% of each sample had been saved as input). Magnetic beads (Dynabeads, Thermo Fisher Scientific) were washed with 1 ml lysis buffer and DG cell lysate/Ab mix was added to the beads, followed by incubation for 4 hr at 4°C on a rotating wheel. Beads were washed five times with activated lysis buffer and resuspended in milli-Q water.

## Immunoblot analysis

5× Laemmli buffer was added to the samples to reach a final volume of 1× and heated 5 min at 95°C at 750 rpm. Protein samples were separated on 10% SDS-polyacrylamide gels and transferred to PVDF membrane (Immobilon). The membrane was blocked for 1 hr with 5% BSA in TBS-T followed by overnight incubation at 4°C with antibody (anti-Drosha Cell Signaling, rabbit, 1:1000 in 5 %BSA; anti-SAFB, Abcam, rabbit, 1:1000, 2.5% BSA; anti-PKC-alpha [H-7] Santa Cruz, mouse, 1:500 1% BSA). Membrane was washed three times for 5 min with TBS-T followed by secondary antibody incubation (anti-rabbit IgG light chain Jackson ImmunoResearch Labs, 1:5000) for 1 hr at room temperature. Membrane was washed three times with TBS-T and once with TBS. Bands were detected by chemiluminescence (ECL, GE Healthcare).

## Affinity purification and sample preparation for MS-based proteome analysis

IP and pull-down probes of DG NSCs were subjected to on-bead digestion (*Hubner et al., 2010*) by trypsin (5 µg/ml, Promega) in 1.6 M urea/0.1 M ammonium bicarbonate buffer at 27°C for 30 min. Supernatant eluates containing active trypsin were further incubated with 1 mM TCEP at room temperature overnight. Carbamidomethylation of cysteines was performed next using 5 mM chloroacetamide in the dark for 30 min. The tryptic digest was acidified (pH <3) using TFA and desalted using C18 reversed phase spin columns (Harvard Apparatus) according to the protocol of the manufacturer. Dried peptides were dissolved in 0.1% aqueous formic acid solution at a concentration of 0.2 mg/ml prior to injection into the mass spectrometer.

## Mass spectrometry analysis

For each sample, aliquots of 0.4 µg of total peptides were subjected to LC-MS analysis using a dual pressure LTQ-Orbitrap Elite mass spectrometer connected to an electrospray ion source (both Thermo Fisher Scientific) and a custom-made column heater set to 60°C. Peptide separation was carried out using an EASY nLC-1000 system (Thermo Fisher Scientific) equipped with an RP-HPLC column (75 µm×30 cm) packed in-house with C18 resin (ReproSil-Pur C18-AQ, 1.9 µm resin; Dr. Maisch

GmbH, Germany) using a linear gradient from 95% solvent A (0.1% formic acid in water) and 5% solvent B (80% acetonitrile, 0.1% formic acid, in water) to 35% solvent B over 50 min to 50% solvent B over 10 min to 95% solvent B over 2 min and 95% solvent B over 18 min at a flow rate of 0.2 µl/min. The data acquisition mode was set to obtain one high-resolution MS scan in the FT part of the mass spectrometer at a resolution of 240,000 full width at half maximum (at 400 m/z, MS1) followed by MS$^2$ scans in the linear ion trap of the 20 most intense MS signals. The charged state screening modus was enabled to exclude unassigned and singly charged ions and the dynamic exclusion duration was set to 30 s. The collision energy was set to 35%, and one microscan was acquired for each spectrum.

## Protein identification and label-free quantification

The acquired raw-files were imported into the Progenesis QI software (v2.0, Nonlinear Dynamics Limited), which was used to extract peptide precursor ion intensities across all samples applying the default parameters. The generated mgf-files were searched using MASCOT against a decoy database containing normal and reverse sequences of the *Mus musculus* proteome (UniProt, April 2017) and commonly observed contaminants (in total 34,490 sequences) generated using the SequenceReverser tool from the MaxQuant software (v1.0.13.13). The following search criteria were used: full tryptic specificity was required (cleavage after lysine or arginine residues, unless followed by proline); three missed cleavages were allowed; carbamidomethylation (C) was set as fixed modification; oxidation (M) and protein N-terminal acetylation were applied as variable modifications; mass tolerance of 10 ppm (precursor) and 0.6 Da (fragments). The database search results were filtered using the ion score to set the FDR to 1% on the peptide and protein level, respectively, based on the number of reverse protein sequence hits in the datasets.

Quantitative analysis results from label-free quantification were normalized and statically analyzed using the SafeQuant R package v.2.3.4 (*Ahrné, 2020*; *Ahrné et al., 2016*) to obtain protein relative abundances. This analysis included global data normalization by equalizing the total peak/reporter areas across all LC-MS runs, summation of peak areas per protein, and LC-MS/MS run, followed by calculation of protein abundance ratios. Only isoform-specific peptide ion signals were considered for quantification. The summarized protein expression values were used for statistical testing of between condition differentially abundant proteins. Here, empirical Bayes moderated t-tests were applied, as implemented in the R/Bioconductor limma package (*Smyth, 2015*; *Ritchie et al., 2015*). The resulting per protein and condition comparison p-values were adjusted for multiple testing using the Benjamini-Hochberg method. All LC-MS analysis runs are acquired from independent biological samples. To meet additional assumptions (normality and homoscedasticity) underlying the use of linear regression models and Student's t-test, MS-intensity signals are transformed from the linear to the log-scale. Unless stated otherwise linear regression was performed using the ordinary least square method as implemented in *base* package of R v.3.1.2 (http://www.R-project.org/).

The sample size of three biological replicates was chosen assuming a within-group MS-signal coefficient of variation of 10%. When applying a two-sample, two-sided Student's t-test this gives adequate power (80%) to detect protein abundance fold changes higher than 1.65, per statistical test. Note that the statistical package used to assess protein abundance changes, SafeQuant, employs a moderated t-test, which has been shown to provide higher power than the Student's t-test. We did not do any simulations to assess power, upon correction for multiple testing (Benjamini-Hochberg correction), as a function of different effect sizes and assumed proportions of differentially abundant proteins.

## RNA biotinylation

*Nfib* 3' UTR and 5' UTR HP-forming regions were in vitro transcribed using T7 transcriptase (NEB, E2040) and purified with Trizol extraction (described in RT-qPCR paragraph). RNA was biotinylated using the Pierce RNA 3' End Desthiobiotinylation Kit (Thermo Fisher Scientific). 50 pmol of RNA was incubated in T4 RNA ligase buffer at 16°C overnight. RNA ligase was extracted with 100 µl chloroform:isoamyl alcohol (Sigma-Aldrich, C0549) and centrifuged 2 min at maximum speed. Aqueous phase was precipitated overnight in 5 M NaCl. Sample was centrifuged at 13,000 × *g* for 20 min at 4°C. Pellet was washed with 70% ethanol and resuspended in dH$_2$O. Labeling efficiency was determined by dot blotting using the chemiluminescent nucleic acid detection module kit (89880, Thermo Fisher Scientific).

## RNA pull-down of NSC proteins

Experiment was performed as described in the manufacturer's protocol (Pierce Magnetic RNA-Protein Pull-Down Kit, Thermo Fisher Scientific, 20164). 50 µl of streptavidin magnetic beads were washed three times with 20 mM Tris, followed by 1× RNA capture buffer. 50 pmol of previously desthiobiotinylated RNA was added to the beads and incubated for 30 min on a heated shaker plate at 24°C and 750 rpm. Beads were washed three times with 20 mM Tris and 100 µl 1× Protein-RNA-binding buffer, followed by 100 µl of RNA protein-binding buffer including 60 µl of protein lysate (2 mg/ml) (cell lysis described in corresponding paragraph) for 60 min 4°C with agitation. The beads were washed three times with 100 µl of the kit 1× wash buffer and then further processed for WB or MS[2].

## Enrichment analysis and candidate selection

Datasets of significantly enriched proteins were analyzed for process networks by MetaCore (Cortellis) and molecular processes by GO terms (PANTHER, GENEONTOLOGY). An interaction network was drawn using STRING database (ELIXIR). Percental enrichment in MetaCore bar plot was calculated as listed in corresponding category out of total dataset. Significance is determined by p-value. For STRING network analysis, only experimentally determined data and curated databases were considered. Nodes with ≥1 Edge are displayed. Network was visualized and analyzed with Cytoscape software (*Shannon et al., 2003*). Prime candidates for functional analysis in reporter assay were selected using the following criteria: (i) enrichment in MS datasets; (ii) relevance in MetaCore, GO, and STRING analysis; (iii) Drosha interactions reported in the literature.

## Tandem RNA immunoprecipitation

We developed RIP[2] based on the protocol of the commercial kit for sequential chromatin immunoprecipitation (Re-ChIP-IT, Active Motif, #53016) but added RNase inhibitors (RNaseIN) to all steps and reaction mixtures.

Cultured NSCs were fixed with 1% paraformaldehyde for 10 min before being lysed in Re-ChIP-IT lysis buffer and DNA sheared by five cycles of sonication (Bioruptor, 30 s on/30 s off, 320 W) at 4°C. After sonication, the lysates appeared homogeneous, were centrifuged at 14,000 × *g* for 10 min, and the supernatants collected. 10 µl of the lysates were kept as 'input' samples and stored at –20°C. An IP reaction mix for the first RIP was prepared according to the vendor's protocol, using primary antibodies (anti-Drosha or rabbit IgG control). The first RIP was incubated overnight at 4°C on a rotating wheel. The samples were washed according to the vendor's protocol (Re-ChIP-IT, Active Motif, #53016). Sample elution was carried out for 30 min at room temperature on a rotating wheel using Re-ChIP-IT Elution Buffer (Active Motif, #53016). The samples were desalted and 10 µl samples were saved as 'post 1st RIP' or simply 'RIP' (as in *Figure 6J*) samples for both IgG and anti-Drosha.

For the second IP, an RIP reaction mix was prepared using 90 µl of the eluate from the first RIP, using anti-SAFB or rabbit IgG antibodies. The second RIPs were incubated overnight at 4°C on a rotating wheel. The samples were washed according to the manufacturer's instructions (Active Motif, #53016). Bound protein-RNA complexes were eluted using the Elution Buffer AM2 (Active Motif, #53016). Samples were treated with Proteinase K to digest proteins and RNases. RNA extraction was performed with Trizol, chloroform, and isopropanol and RNA precipitates washed with 70% ethanol. The RNA pellet was air-dried and resuspended in RNAse-free water. The samples were treated with DNase I to remove any remaining gDNA and the DNase inactivated by incubation for 10 min at 65°C. The remaining RNA was subjected to reverse transcription with SuperScript IV (Thermo Fisher Scientific, Cat #11766500) and random hexamer primers. The cDNA was used for PCR using *Nfib* 3'-f: ( GTTACCTCTGCATGCAACAG) and *Nfib* 3'-r: (GCTGCAGCTAAGCCAACCT) primers and amplicons separated on a 1% agarose gels by electrophoresis.

## Stable NSC line generation

*Nfib* 3' UTR, 5' UTR, and EGFPd2 sequences were cloned in the multiple cloning site of pTet-One vector (Takara-Clontech, 634301). DG NSC *Drosha*^fl/fl^ cells were brought in suspension by incubating with 0.25% trypsin (Gibco #15090) in Versene (Gibco #15040) for 5 min at 37°C followed by centrifugation at 80 × *g* for 5 min at room temperature. Cells were electroporated with the corresponding DNA vector using a 4D-Nucleofector (Lonza, program DS-112) according to the protocol of manufacturer and re-plated in plastic dish (Costar) coated with 100 µg/ml poly-L-lysine (Sigma-Aldrich) and

1 µg/ml laminin (Sigma-Aldrich). Cells were induced with 1 µg/ml doxycycline in DG NSC medium. 48 hr post-induction, cells were sorted for GFP⁺ by flow cytometry (FACSaria III, BD Biosciences). All GFP⁺ cells were collected, centrifuged at 80 × $g$ for 5 min, and resuspended in DG NSC medium. Cells were re-plated and passaged three times. Thereafter, cells were re-induced with 1 µg/ml doxycycline and re-sorted twice. Correct genotype of cell line was confirmed by genotyping using *Nfib* UTR and EGFPd2-specific primers.

## RBP overexpression vector generation

Coding sequences of selected prime candidates were cloned in a CAG::IRES-Cfp expression vector, using the In-Fusion cloning kit (Takara) following the manufacturer's protocols. Protein cDNAs from source vector were amplified by PCR and cloned upstream of the IRES-Cfp sequences in the CAG::IRES-Cfp vector. Sequences of all resulting vectors were verified by Sanger sequencing (Eurofins).

## Tet-on DG NSC nucleofection and FACS analysis

Tet-on ctrl, Tet-on 3' UTR HP, and 5' UTR HP DG NSCs were brought in suspension by incubating with 0.25% trypsin (Gibco #15090) in Versene (Gibco #15040) for 4 min at 37°C followed by centrifugation at 80 × $g$ for 5 min at room temperature. Cells were electroporated with either CAG::IRES-Cfp or CAG::Rbp-IRES-Cfp vector (5 µg) using a 4D-Nucleofector (Lonza, program DS-112) in 16-well stripes (500,000 cells/well) and re-plated in plastic dish (Costar) coated with 100 µg/ml poly-L-lysine (Sigma-Aldrich) and 1 µg/ml laminin (Sigma-Aldrich). For rescue experiments, Tet-on ctrl, Tet-on 3' UTR HP, and 5' UTR HP *Drosha*^fl/fl^ DG NSCs were electroporated with 1.6 µg of *Cre-IRES-Tomato* expression vector (pMITom::nlsCre) and 3.4 µg of CAG::IRES-Cfp or CAG::Safb-IRES-Cfp by 4D-Nucleofector (program DS-112).

After 24 or 48 hr, cells were induced with 1 µg/ml doxycycline in DG NSC medium. 48 hr post-induction, cells were brought in suspension with 0.25% trypsin in Versene collected in DMEM/F12 without red phenol, filtered through a 40 µm cell sieve (Miltenyi Biotec) and analyzed by flow cytometry (FACSaria III, BD Biosciences). GFP- and CFP-double negative, GFP-single positive and CFP-single positive DG NSCs were used to create the compensation matrix and set the sorting gates. At least 100,000 events were recorded for each sample and MFI (median fluorescence intensity) of the population of interest was subsequently analyzed with FlowJo (Becton Dickinson). GFP⁺, CFP⁺GFP⁺, Tom⁺ cells were sorted, centrifuged at 80 × $g$ for 5 min, and used for RNA isolation and gene expression analysis (see below).

## SAFB-ΔRE-HA mutant generation and overexpression

The SAFB-ΔRE-HA mutant was generated using the Q5 Site-Directed Mutagenesis Kit (NEB). 5'-Overlapping primers were designed and the CAG::Safb-HA-IRES-Cfp vector was amplified excluding the RE region (amino acids 552–708). The vector was re-ligated, transformed, and endotoxin-free maxiprep (QIAGEN) DNA sequenced. The SAFB-ΔRE mutant variant encoding cDNA was cloned into the same CAG::IRES-Cfp expression used to express the RBPs in the RBP overexpression analysis (see RBP overexpression vector generation). 2 million N2A cells were transfected with 10 µg vector DNA using TransFectin (Bio-Rad) with either CAG::IRES-Cfp ctrl., CAG::Safb-HA-IRES-Cfp, or CAG::Safb-ΔRE-HA-IRES-Cfp. 48 hr after transfection the cells were prepared for co-IP experiments (see Cell lysis for protemics and Drosha co-IP assay). Tet-on ctrl and Tet-on 3' UTR HP DG NSCs were electroporated with either CAG::IRES-Cfp ctrl., CAG::Safb-HA-IRES-Cfp, or CAG::Safb-ΔRE-HA-IRES-Cfp and analyzed by FACS as described above (see Tet-on DG NSC nucleofection and FACS analysis).

## Crosslinking immunoprecipitation

WT mouse DG NSCs in culture were washed with cold PBS and crosslinked at 254 nm, 300 mJ/cm² in a BioLink UV-Crosslinker. Cells were lysed with RIPA buffer (0.1 M sodium phosphate pH 7.2, 150 mM sodium chloride, 0.1% SDS, 1% sodium deoxycholate, 1% NP-40, 1 mM activated $Na_3VO_4$, 1 mM NaF, 1× cOmplete Protease Inhibitor [Roche]). Lysate was collected and centrifuged for 5 min at 13,000 × $g$ at 4°C. Sample was treated with 10 µl DNase (Roche) and incubated for 5 min in a heated shaker at 37°C and 1000 rpm, followed by centrifugation at 13,000 × $g$ at 4°C and probe separation. 50 µl magnetic beads (Dynabeads, Invitrogen) were washed with lysis buffer and incubated with target specific antibody (anti-SAFB, Abcam, rabbit, 1:50) for 1 hr at room temperature on a rotating wheel.

Beads were washed twice with lysis buffer and incubated with crosslinked NSC lysate for 2 hr at 4°C on a rotating wheel. Beads were washed five times with lysis buffer, incubated with 4 mg/ml Proteinase K (Roche) in Proteinase K buffer (100 mM Tris HCl pH 7.5, 50 mM NaCl, 10 mM EDTA) for 20 min at 37°C at 1000 rpm, followed by downstream RNA analysis. PCR primers for the genes of interest were designed by the Universal Probe Library Assay Design Center (Roche) (primer sequences listed in Key resources table).

## RNA isolation and RT-qPCR
Total RNA was isolated following the standard phenol-chloroform protocol from the manufacturer (Trizol, Life Technologies). 100% Trizol was added to the sample, followed by 20% of total volume chloroform. Sample was centrifuged at 12,000 × $g$ for 15 min at 4°C. Aqueous phase was extracted and RNA was precipitated with isopropanol and washed with ethanol. RNA pellet was resuspended in RNase-free Milli-Q water. Reverse transcription was performed using Superscript IV Vilo following the manufacturer's protocol (Thermo Fisher Scientific, 11766050). RNA was incubated for 5 min with ezDNAse enzyme at 37°C, followed by incubation with SuperScript IV VILO Master Mix at 25°C for 10 min, 50°C for 10 min, and 85°C for 5 min. For expression analysis of genes of interest, we used the comparative Ct method using the *Rpl13a* and *Actin* mRNA as normalizing genes. Experiments were performed using a qTOWER$^3$ real-time PCR machine (Analytik Jena). Three biological replicates for each genotype and three technical replicates for each gene were analyzed.

## esiRNA-mediated *Safb* RNA KD
WT DG NSCs were dissociated with 0.25% trypsin in Versene followed by centrifugation at 80 × $g$ for 5 min at room temperature. Cells were electroporated with either 70 pmol esiRNA Luciferase or esiRNA *Safb* (Sigma-Aldrich) in combination with 30 pmol RNA probe Alexa555 (Sigma-Aldrich) using a 4D-Nucleofector (Lonza, program DS-112) in 16-well stripes (500,000 cells/well) and re-plated on glass coverslips coated with 100 µg/ml poly-L-lysine (Sigma-Aldrich) and 1 µg/ml laminin (Sigma-Aldrich). Cells were fixed 48, 72, and 96 hr after nucleofection in 4% paraformaldehyde in 0.1 M phosphate buffer for 10 min. N2A cells were transfected with Lipofectamine RNAiMAX (Thermo Fisher Scientific) according to the manufacturer's instructions. N2A cells were transfected with 90 pmol of esiRNA Luciferase or esiRNA *Safb* (Sigma-Aldrich) in a six-well plate, seeded 1 day before transfection (250,000 cells/well in DMEM 4.5 g/l, 10% fetal bovine serum). 48 hr after transfection, cells were lysed into RIPA buffer for downstream analysis, followed by sonication for five cycles (Bioruptor, 30 s on/30 s off, 320 W) at 4°C. Lysates were centrifuged at 13,000 × $g$ for 10 min at 4°C. The supernatants were transferred to a new tube and protein quantified by BCA assay (Thermo Fisher Scientific, #23250) according to the manufacturer's instructions.

## Analysis of SAFB regulation of Drosha activity by in vitro processing assay
N2A cells were transfected with esiRNA *Safb* as described above (esiRNA-mediated *Safb* RNA KD). After 48 hr, samples were lysed and the full lysate was treated with RNase A and DNAse I and immunoprecipitated for Drosha as described above (see Cell lysis for protemics and Drosha co-IP assay). For the in vitro processing assay (*Rolando et al., 2016*), samples were mixed in the following ratios: Sample 1 (15 µl WT), Sample 2 (10 µl WT, 5 µl KD), Sample 3 (5 µl WT, 10 µl KD), Sample 4 (15 µl KD). A total of 30 µl of the processing reaction were prepared and contained: 15 µl of beads from Drosha immunoprecipitated beads, mixed or control fraction, 6.4 mM MgCl$_2$, 0.75 µl RNaseIN (Invitrogen), and 0.5 µg RNA probe containing T7 RNA polymerase (NEB) transcribed *Nfib* 3' UTR HP RNA or AR RNA/*Nfib* 3' UTR RNA hybrid RNA as control. The reaction was carried out at 25°C for 30 min. RNA was extracted using Trizol and subsequently analyzed by RT-qPCR.

## GFP reporter analysis after glial differentiation
Tet-on ctrl and Tet-on 3' UTR HP DG NSCs were cultured under expansion conditions until 70–80% confluent and then astrocytic differentiation induced by removal of growth factors EGF and FGF and treatment with 10% fetal bovine serum. After 5 days, the cells were induced with 1 µg/ml doxycycline. 48 hr post-doxycycline induction, the cells were dissociated with 0.25% trypsin in Versene, collected in DMEM/F12 without red phenol, stained with BV421-labeled anti-CD44 (BioLegend,

mouse, 1:10) or BV421-labeled rat IgG2b isotype control (BioLegend, mouse, 1:10) in 0.05% BSA for 20 min on ice protected from light, filtered through a 40 μm cell sieve (Miltenyi Biotec) and analyzed by flow cytometry (FACS Aria III, BD Biosciences). GFP- and CD44-double negative, GFP-single positive, and CD44-single positive NSCs were used to create the compensation matrix and set the sorting gates. At least 100,000 events were recorded for each sample and MFI of the population of interest was subsequently analyzed with FlowJo (Becton Dickinson).

## SAFB overexpression in postnatal V-SVZ NSCs

Postnatal C57BL6 V-SVZ-derived neurospheres were prepared as described previously (*Giachino et al., 2009*). Postnatal mice were sacrificed, decapitated, and their brains collected in 6 cm dishes containing sterile cold HBSS. The meninges were carefully removed and the striatum isolated by dissection. The lateral walls of the striatum were dissected and the tissue was dissociated in 500 μl pre-warmed Papain solution at 37°C for 30 min followed by addition of 500 μl of pre-warmed Trypsin inhibitor and incubation for a further 5 min at 37°C. The tissue was mechanically dissociated with 1 ml and 200 μl pipette tips. After addition of 9 ml of DMEM/F12, the samples were centrifuged at 80 × *g* for 5 min to remove debris. The cell pellet from each animal was resuspended in neurosphere medium DMEM/F12 (Gibco, Invitrogen), 2% B27 (Gibco, Invitrogen), EGF 10 ng/ml (R&D Systems) and plated in T25 flasks. The cells were fed every 2 days with fresh neurosphere medium. The cells were passaged every 4 days.

Dissociated V-SVZ neurosphere cells were electroporated with either CAG::IRES-Cfp or CAG::Safb-IRES-Cfp expression vectors (4 μg) using a 4D-Nucleofector (Lonza, program DS-112) in 16-well stripes (500,000 cells/well) and re-plated onto 13 mm glass coverslips in 24-well plates coated with 100 μg/ml poly-L-lysine (Sigma-Aldrich) and 1 μg/ml laminin (Sigma-Aldrich) in neurosphere medium. After 48 hr differentiation was induced by replacing the medium with neurosphere medium without EGF and adding 10% fetal bovine serum. The cells were differentiated for a further 48 hr and then fixed for immunocytochemical analysis by 10 min incubation in 4% paraformaldehyde in 0.1 M phosphate buffer. The cells were then stained using antibodies against anti-GFP (Aves Labs, chicken, 1:500), anti-SAFB (Abcam, rabbit, 1:300), and anti-SOX10 (Santa Cruz, goat, 1:200) antibodies (see Key resources table). The experiment was repeated three times with three biological replicates each.

## Quantification and statistical analysis

Images of immunostainings were captured and processed on a Confocal Leica SP5 and Apotome2 (Zeiss), as well as processed in Fiji. According to the Swiss governmental guidelines and requirements, the principles of the 3Rs for animal research were taking into consideration to reduce the number of mice used in the experiment. Data are presented as averages of indicated number of samples. Data representation and statistical analysis were performed using GraphPad Prism software. Statistical comparisons were conducted by two-tailed unpaired Student's t test, Mann-Whitney test, or one-way ANOVA test, as indicated. The size of samples (**n**) is described in the figure legends. Error bars are SEM.

For SAFB motif analysis, the top 20 of pentamers (ranked for the highest Z-scores) identified in previous SAFB CLIP experiments (*Rivers et al., 2015*) were mapped on the *Nfib* 3'UTR sequence. These most enriched pentamers were also used to create the consensus binding motif with the web-based WebLogo software (http://weblogo.berkeley.edu/).

## Acknowledgements

We thank the members of the VT laboratory for helpful discussions and Frank Sager for excellent technical assistance. We thank the Proteomics Facility of the Biozentrum Basel and the Animal Core Facility of the University of Basel. This work was supported by the Swiss National Science Foundation (31003A_162609 and 31003A_182388).

## Additional information

### Funding

| Funder | Grant reference number | Author |
|--------|------------------------|--------|
| Schweizerischer Nationalfonds zur Förderung der Wissenschaftlichen Forschung | 31003A_162609 | Verdon Taylor |
| Schweizerischer Nationalfonds zur Förderung der Wissenschaftlichen Forschung | 31003A_182388 | Verdon Taylor |

The funders had no role in study design, data collection and interpretation, or the decision to submit the work for publication.

### Author contributions

Pascal Forcella, Niklas Ifflander, Elli-Anna Balta, Aikaterini Lampada, Data curation, Formal analysis, Investigation, Writing - original draft, Writing - review and editing; Chiara Rolando, Tanzila Mukhtar, Conceptualization, Data curation, Formal analysis, Validation, Investigation, Writing - original draft, Writing - review and editing; Claudio Giachino, Formal analysis, Investigation, Writing - original draft, Writing - review and editing; Thomas Bock, Data curation, Formal analysis, Investigation, Writing - review and editing; Verdon Taylor, Conceptualization, Supervision, Funding acquisition, Writing - original draft, Project administration, Writing - review and editing

### Author ORCIDs

Thomas Bock http://orcid.org/0000-0002-9314-5318
Verdon Taylor http://orcid.org/0000-0003-3497-5976

### Ethics

According to Swiss Federal and Swiss Veterinary office regulations, all mice were bred and kept in a specific pathogen-free animal facility with 12 hours day-night cycle and free access to clean food and water. All mice were healthy and immunocompetent. All procedures were approved by the Basel Cantonal Veterinary Office under license number 2537 (Ethics commission Basel-Stadt, Basel Switzerland).

### Decision letter and Author response

Decision letter https://doi.org/10.7554/eLife.74940.sa1
Author response https://doi.org/10.7554/eLife.74940.sa2

## Additional files

### Supplementary files

• Supplementary file 1. List of proteins identified in the Drosha IP experiments. Proteins significantly enriched over control IP are marked in green.

• Supplementary file 2. Proteomics data and lists of proteins identified in the Nfib mRNA pulldown experiments including non-specific bead interacting proteins and proteins interacting with the Nfib-AR hybrid mRNA probes.

• Supplementary file 3. Interactive iVenn data set comparing proteins identified in the Drosha IP, Nfib 3' UTR HP pull-down and Nfib 5' UTR HP pull-down experiments.

• Source data 1. Excel sheet of quantificantion data and statistical analyses for all figures and figure supplement data.

• Transparent reporting form

## Data availability

The mass spectrometry proteomics data in this manuscript have been deposited to the ProteomeXchange Consortium via the PRIDE (*Perez-Riverol et al., 2019*) partner repository with the accession number PXD017677 and project DOI: https://doi.org/10.6019/PXD017677.

The following dataset was generated:

| Author(s) | Year | Dataset title | Dataset URL | Database and Identifier |
|---|---|---|---|---|
| Taylor V | 2024 | SAFB regulates hippocampal stem cell fate by targeting Drosha to destabilize Nfib mRNA | https://www.ebi.ac.uk/pride/archive/projects/PXD017677 | PRIDE, PXD017677 |

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

# Appendix 1

## Appendix 1—key resources table

| Reagent type (species) or resource | Designation | Source or reference | Identifiers | Additional information |
|---|---|---|---|---|
| Antibody | Anti-BLBP (rabbit polyclonal) | Millipore | Cat #ABN14, RRID:AB_10000325 | IF: 1:500 |
| Antibody | Anti-activated cleaved Caspase-3 (rabbit monoclonal) | Cell Signaling Technology | Cat #9664, RRID:AB_2070042 | IF: 1:400 |
| Antibody | Anti-goat Alexa647 (donkey polyclonal) | Jackson ImmunoResearch | Cat #705-605-147, RRID:AB_2340437 | IF: 1:600 |
| Antibody | Anti-mouse Cy3 (donkey polyclonal) | Jackson ImmunoResearch | Cat #715-165-151, RRID:AB_2315777 | IF: 1:600 |
| Antibody | Anti-rabbit Alexa488 (donkey polyclonal) | Jackson ImmunoResearch | Cat #711-545-152, RRID:AB_2313584 | IF: 1:600 |
| Antibody | BV421-labeled anti-CD44 (mouse monoclonal) | BioLegend | Cat #103039, RRID:AB_10895752 | FACS: 1:10 |
| Antibody | BV421-labeled anti-rat IgG2b isotype control (mouse monoclonal) | BioLegend | Cat #400639, RRID:AB_10895758 | FACS: 1:10 |
| Antibody | Anti-Drosha (rabbit monoclonal) | Cell Signaling Technology | Cat #3364, RRID:AB_2238644 | IB: 1:1000 IP: 1:50 |
| Antibody | IgG control (rabbit monoclonal) | Cell Signaling Technology | Cat #3900, RRID:AB_1550038 | IP: 1:50 |
| Antibody | Anti-HA tag (rabbit monoclonal) | Cell Signaling Technology | Cat #3724, RRID:AB_1549585 | IB: 1:1000 |
| Antibody | Anti-GAPDH (mouse monoclonal) | Calbiochem | CB1001, RRID:AB_2107426 | IB: 1:1500 |
| Antibody | Anti-GFAP (Glial Fibrillary acidic protein) (mouse monoclonal) | Sigma-Aldrich | Cat #G3893, RRID:AB_477010 | IF: 1:200 |
| Antibody | Anti-GFP (Green fluorescent protein) (sheep polyclonal) | AbD Serotec/Bio-Rad | Cat #4745–1051 RRID:AB_619712 | IF: 1:250 |
| Antibody | Anti-GFP (Green fluorescent protein) (rabbit polyclonal) | Invitrogen/Thermo Fisher Scientific | Cat #A11122 RRID:AB_221569 | IF: 1:700 |
| Antibody | Anti-GFP (Green fluorescent protein) (chicken) | Aves Labs | Cat #GFP-1020, RRID:AB_10000240 | IF: 1:500 |
| Antibody | Anti-MAP2 (mouse monoclonal) | Sigma-Aldrich | Cat #M4403, RRID:AB_477193 | IF: 1:200 |
| Antibody | Anti-NFIB (rabbit polyclonal) | Invitrogen/Thermo Fisher Scientific | Cat #PA5-52032, RRID:AB_2644645 | IF: 1:1000 |
| Antibody | Anti-OLIG2 (rabbit polyclonal) | Chemicon | Cat #AB9610, RRID:AB_570666 | IF: 1:500 |
| Antibody | Anti-PKC-alpha (mouse monoclonal) | Santa Cruz | Cat #sc-8393, RRID:AB_628142 | IB: 1:500 |
| Antibody | HRP-conjugated anti-Rabbit IgG, light chain (mouse monoclonal) | Jackson ImmunoResearch Labs | Cat #211-032-171, RRID:AB_2339149 | IB: 1:5000 |
| Antibody | Anti-S100b (mouse monoclonal) | Sigma-Aldrich | Cat #S2532, RRID:AB_477499 | IF: 1:100 |
| Antibody | Anti-SAFB (rabbit monoclonal) | Abcam | Cat #ab187650, RRID:AB_2814774 | IF: 1:200 IP: 1:50 |
| Antibody | Anti-SOX10 (goat polyclonal) | Santa Cruz | Cat #sc-17342; RRID:AB_2195374 | IF: 1:500 |

*Appendix 1 Continued on next page*

*Appendix 1 Continued*

| Reagent type (species) or resource | Designation | Source or reference | Identifiers | Additional information |
|---|---|---|---|---|
| Chemical compound, drug | B27 Supplement (50×) | Gibco | Cat #17504044 | |
| Chemical compound, drug | DMEM:F-12+GlutaMAX | Gibco | Cat #31331-028 | |
| Chemical compound, drug | cOmplete Protease Inhibitor Cocktail | Roche/Sigma-Aldrich | Cat #11697498001 | |
| Chemical compound, drug | EGF (Recombinant Mouse EGF Protein, CF) | R&D Systems | Cat #MAB2028-100 | |
| Chemical compound, drug | FGF2 (Recombinant Mouse FGF basic, CF) | R&D Systems | Cat #233-FB-025 | |
| Chemical compound, drug | Formaldehyde (16% methanol free) | Thermo Fisher Scientific | Cat #28906 | |
| Chemical compound, drug | Glycine | Sigma-Aldrich | Cat #50046-1KG | |
| Chemical compound, drug | L-15 Medium | Invitrogen | Cat #31415029 | |
| Chemical compound, drug | Normal Donkey Serum | Jackson ImmunoResearch | Cat #017-000-121 | |
| Chemical compound, drug | Papain | Sigma-Aldrich | Cat #P3125-100MG | |
| Chemical compound, drug | Protein G Dynabeads | Invitrogen | Cat #10003D | |
| Chemical compound, drug | SsoAdvanced Universal Probes Supermix | Bio-Rad | Cat #172-5280 | |
| Chemical compound, drug | Trizol | Invitrogen | Cat #VX15596026 | |
| Chemical compound, drug | Trypsin inhibitor Glycine max (Soybean)/Ovomucoid | Sigma-Aldrich | Cat #T6522-100MG | |
| Commercial assay or kit | ECL Blotting Reagents | GE Healthcare | Cat #GERPN2109 | |
| Commercial assay or kit | HiScribe T7 High Yield RNA Synthesis Kit | New England Biolabs | Cat #E2040 | |
| Commercial assay or kit | In-Fusion HD Cloning Plus | Takara-Clontech | Cat #638910 | |
| Commercial assay or kit | Pierce BCA Protein Assay Kit | Thermo Fisher Scientific | Cat #23225 | |
| Commercial assay or kit | Pierce Magnetic RNA-Protein Pull-Down Kit | Thermo Fisher Scientific | Cat #20164 | |
| Commercial assay or kit | Pierce RNA 3' End Desthiobiotinylation Kit | Thermo Fisher Scientific | Cat #20163 | |
| Commercial assay or kit | P3 Primary Cell 4D-Nucleofector X Kit S | Lonza | Cat #V4XP-3032 | |
| Commercial assay or kit | Q5 Site-Directed Mutagenesis Kit | New England Biolabs | Cat #E0554S | |
| Commercial assay or kit | Lipofectamine RNAiMAX Transfection Reagent | Thermo Fisher Scientific | Cat #13778100 | |

*Appendix 1 Continued on next page*

*Appendix 1 Continued*

| Reagent type (species) or resource | Designation | Source or reference | Identifiers | Additional information |
|---|---|---|---|---|
| Commercial assay or kit | SuperScript IV VILO Master Mix with ezDNase Enzyme | Thermo Fisher Scientific | Cat #11766500 | |
| Commercial assay or kit | TransFectin Lipid Reagent | Bio-Rad | Cat #1703351 | |
| Commercial assay or kit | Re-ChIP-IT | Active Motif | Cat #53016 | |
| Genetic reagent (*Mouse*) | Rosa26-CAG::Egfp | *Lugert et al., 2012* | Gt(ROSA)26Sor$^{tm1(CAG-EGFP)}$ | C57BL6 genetic background |
| Genetic reagent (*Mouse*) | Hes5::CreER$^{T2}$ | *Lugert et al., 2010* | Tg(Hes5-cre/ERT2)$^{2Vtlr}$ | C57BL6 genetic background |
| Genetic reagent (*Mouse*) | Rosa26-CAG::Egfp Hes5::CreER$^{T2}$ | This paper | Gt(ROSA)26Sor$^{tm1(CAG-EGFP)}$ Tg(Hes5-cre/ERT2)$^{2Vtlr}$ | C57BL6 genetic background |
| Genetic reagent (*Mouse*) | *Drosha*$^{fl/fl}$ | *Chong et al., 2010* | Drosha$^{tm1Litt}$ | C57BL6 genetic background |
| Genetic reagent (*Mouse*) | Rosa26-CAG::Egfp Hes5::CreER$^{T2}$ *Drosha*$^{fl/fl}$ | This paper | Gt(ROSA)26Sor$^{tm1(CAG-EGFP)}$ Tg(Hes5-cre/ERT2)$^{2Vtlr}$ Drosha$^{tm1Litt}$ | C57BL6 genetic background |
| Genetic reagent (*Mouse*) | *Safb*$^{fl/fl}$ | This paper | Safb$^{em1(flE2)Vtlr}$ | C57BL6 genetic background |
| Genetic reagent (*Mouse*) | Rosa26-CAG::Egfp Hes5::CreER$^{T2}$ *Safb*$^{fl/fl}$ | This paper | Gt(ROSA)26Sor$^{tm1(CAG-EGFP)}$ Tg(Hes5-cre/ERT2)$^{2Vtlr}$ Safb$^{em1(flE2)Vtlr}$ | C57BL6 genetic background |
| Biological sample (*Mouse*) | Adult hippocampal WT neural stem cells | This paper | WT DG NSCs | Primary neural stem cells from C57BL6 mice |
| Biological sample (*Mouse*) | Adult hippocampal *Drosha*$^{fl/fl}$ Tet-on EGFPd2 neural stem cells | This paper | Tet-on ctrl DG NSCs | Primary neural stem cells from C57BL6 mice |
| Biological sample (*Mouse*) | Adult hippocampal *Drosha*$^{fl/fl}$ Tet-on *Nfib* 3'UTR- EGFPd2 neural stem cells | This paper | Tet-on 3'UTR DG NSCs | Primary neural stem cells from C57BL6 mice |
| Biological sample (*Mouse*) | Adult hippocampal *Drosha*$^{fl/fl}$ Tet-ON *Nfib* 5'UTR- EGFPd2 neural stem cells | This paper | Tet-on 5'UTR DG NSCs | Primary neural stem cells from C57BL6 mice |
| Sequence-based reagent | *Drosha*-f | Roche | N/A | CCGTCTCTAGAAA GGTCCTACAAG (with UPL probe 12) |
| Sequence-based reagent | *Drosha*-r | Roche | N/A | GGCTCAGGAGC AACTGGTAA (with UPL probe 12) |
| Sequence-based reagent | Egfpd2-f | Roche | N/A | GAAGCGCGA TCACATGGT (with UPL probe 67) |
| Sequence-based reagent | Egfpd2-r | Roche | N/A | CCATGCCGAG AGTGATCC (with UPL probe 67) |
| Sequence-based reagent | *HNRNPU*-f | Roche | N/A | CAACAGAGGGAA CTATAACCAGAAC (with UPL probe 74) |
| Sequence-based reagent | *HNRNPU*-r | Roche | N/A | GCTTCTGACC CCAGAATTGA (with UPL probe 74) |
| Sequence-based reagent | MISSION esiRNA *Safb* | Sigma-Aldrich | Cat #EMU167381 | esiRNA |
| Sequence-based reagent | MISSION esiRNA *rLuciferase* | Sigma-Aldrich | Cat #EHURLUC | esiRNA |

*Appendix 1 Continued*

| Reagent type (species) or resource | Designation | Source or reference | Identifiers | Additional information |
|---|---|---|---|---|
| Sequence-based reagent | BLOCK-iT Alexa Fluor 555 | Sigma-Aldrich | Cat #14750100 | esiRNA control |
| Sequence-based reagent | *Nfib*-f | Roche | N/A | TCAAGCCAATC GATATGTGG (with UPL probe 4) |
| Sequence-based reagent | *Nfib*-r | Roche | N/A | GAACCAAGCTA GCCCAGGTA (with UPL probe 4) |
| Sequence-based reagent | *Nfib* 3'-f | This paper | N/A | GTTACCTCTGC ATGCAACAG |
| Sequence-based reagent | *Nfib* 3'-r | This paper | N/A | GCTGCAGCTAA GCCAACCT |
| Sequence-based reagent | *UBC*-f | Roche | N/A | GACCAGCAGAG GCTGATCTT (with UPL probe 11) |
| Sequence-based reagent | *UBC*-r | Roche | N/A | CCTCTGAGGCG AAGGACTAA(with UPL probe 11) |
| Sequence-based reagent | *Safb*-f2 | This paper | N/A | GAGAAAGCCTTTGTC GAGGAGCTAGG |
| Sequence-based reagent | *Safb*-r2 | This paper | N/A | GAGGCTATGTGAAG CTGGAAGACCA |
| Recombinant DNA reagent | pSplit2-NO-PRPF6 (plasmid) | *Maita et al., 2014* | RRID: Addgene_51740 | pSplit2 expression vector |
| Recombinant DNA reagent | pcDNA3.1 SAFB (plasmid) | *Townson et al., 2003* | RRID: Addgene_32742 | pcDNA3.1 expression vector |
| Recombinant DNA reagent | GUM BUB3 (plasmid) | *Toledo et al., 2014* | RRID: Addgene_84029 | expression vector |
| Recombinant DNA reagent | mCherry-PABPN1 (plasmid) | *Chou et al., 2015* | https://doi.org/10.1093/hmg/ddv238 | CMV-based expression vector |
| Recombinant DNA reagent | pCAG TDP-43 (plasmid) | Verdon Taylor Lab | N/A | pCAG expression vector |
| Recombinant DNA reagent | pcDNA3.2-FUS-1-526aa-V5 (plasmid) | *Yoo et al., 2011* | RRID: Addgene_29609 | pcDNA3.2 expression vector |
| Recombinant DNA reagent | pcDNA3 HA Sam68 WT (plasmid) | *Lin et al., 1997* | RRID: Addgene_17690 | pcDNA3 expression vector |
| Recombinant DNA reagent | pCDNA3.1 DGCR8-FLAG (plasmid) | Ueli Suter | N/A | pcDNA3.1 expression vector |
| Recombinant DNA reagent | pcDNA4 DDX5-HA-myc-His (plasmid) | Tanja Vogel Lab | N/A | pcDNA4 expression vector |
| Recombinant DNA reagent | pDESTmycDDX17 (plasmid) | *Landthaler et al., 2008* | RRID: Addgene_19876 | pDEST expression vector |
| Recombinant DNA reagent | pEGFP-DHX9 (plasmid) | *Fidaleo et al., 2015* | https://doi.org/10.18632/oncotarget.5033 | pEGFP expression vector |
| Recombinant DNA reagent | pEGFPC1-6XHis-FLKSRP (plasmid) | *Hall et al., 2004* | RRID: Addgene_23001 | pEGFPC1 expression vector |
| Recombinant DNA reagent | pENTR-D-Topo-hTRIM9 (plasmid) | *Short and Cox, 2006* | RRID: Addgene_51032 | pENTR expression vector |
| Recombinant DNA reagent | pFRT/FLAG/HA-DEST QKI (plasmid) | *Landthaler et al., 2008* | RRID: Addgene_19891 | pFRT expression vector |
| Recombinant DNA reagent | pFRT/TO/HIS/FLAG/HA-HNRNPU (plasmid) | *Baltz et al., 2012* | RRID: Addgene_38068 | pFRT expression vector |
| Recombinant DNA reagent | pFRT/TO/HIS/FLAG/HA-SART1 (plasmid) | *Baltz et al., 2012* | RRID: Addgene_38087 | pFRT expression vector |

*Appendix 1 Continued on next page*

*Appendix 1 Continued*

| Reagent type (species) or resource | Designation | Source or reference | Identifiers | Additional information |
|---|---|---|---|---|
| Recombinant DNA reagent | pHR-HNRNPA1C-mCh-Cry2WT (plasmid) | *Shin et al., 2017* | RRID: Addgene_101226 | pHR expression vector |
| Recombinant DNA reagent | pCAG::[RBP]-IRES-Cfp (plasmid) | This paper | N/A | pCAG expression vector |
| Recombinant DNA reagent | pCAG::IRES-Cfp (plasmid) | *Moore et al., 2015* | N/A | pCAG expression vector |
| Recombinant DNA reagent | pGEMT *Nfib* 3'UTR HP (plasmid) | *Rolando et al., 2016* | https://doi.org/10.1016/j.stem.2016.07.003 | pGEM cloning plasmid |
| Recombinant DNA reagent | pGEMT *Nfib* 5'UTR HP (plasmid) | *Rolando et al., 2016* | https://doi.org/10.1016/j.stem.2016.07.003 | pGEM cloning plasmid |
| Recombinant DNA reagent | pCAG::GFPd2 (plasmid) | *Matsuda and Cepko, 2007* | RRID: Addgene_14760 | pCAG expression vector |
| Recombinant DNA reagent | pMITom::nlsCre (plasmid) | Verdon Taylor Lab | N/A | Retrovirus-based expression vector |
| Recombinant DNA reagent | Tet-on::Egfpd2 (plasmid) | This paper | N/A | pTET-ONE expression vector |
| Recombinant DNA reagent | Tet-on::*Nfib* 3' UTR Egfpd2 (plasmid) | This paper | N/A | pTET-ONE expression vector |
| Recombinant DNA reagent | Tet-on::*Nfib* 5' UTR Egfpd2 (plasmid) | This paper | N/A | pTET-ONE expression vector |
| Recombinant DNA reagent | Tet-One Inducible Expression System (plasmid) | Takara-Clontech | Cat #634301 | pTET-ONE expression vector |
| Software, algorithm | Cytoscape | Cytoscape Consortium | https://cytoscape.org/ | |
| Software, algorithm | FlowJo | Becton, Dickinson & Company | https://www.flowjo.com/ | |
| Software, algorithm | GO PANTHER | GENEONTOLOGY | http://geneontology.org/ | |
| Software, algorithm | GraphPad Prism 8 | GraphPad | https://www.graphpad.com/scientific-software/prism/ | |
| Software, algorithm | Illustrator | Adobe | https://www.adobe.com/Illustrator | |
| Software, algorithm | Lasergene | DNASTAR | https://www.dnastar.com/software/lasergene/ | |
| Software, algorithm | MetaCore | Cortellis | OMICS_02716 | |
| Software, algorithm | Omero | OME | https://www.openmicroscopy.org/about/ | |
| Software, algorithm | Photoshop | Adobe | https://www.adobe.com/Photoshop | |
| Software, algorithm | Fiji (Fiji is just ImageJ) | Open Source | https://fiji.sc/ | |
| Software, algorithm | STRING database | ELIXIR | https://string-db.org/ | |

