## [Editor Report]

The authors provide important information regarding the non-canonical functions of the ribonuclease Drosha in neural stem cell fate determination. They show convincing evidence that Drosha interacts with the Scaffold Attachment Factor B (SAFB) to prevent expression of the transcription factor NFIB, thereby preventing the formation of oligodendrocytes by neural stem cells. Overall their results provide new insight into molecular mechanisms that regulate NSC fate.

---

## [Decision Letter]

**Decision letter after peer review:**

Thank you for submitting your article "Safb1 regulates cell fate determination in adult neural stem cells by enhancing Drosha cleavage of NFIB mRNA" for consideration by *eLife*. Your article has been reviewed by 2 peer reviewers, and the evaluation has been overseen by a Reviewing Editor and Marianne Bronner as the Senior Editor. The reviewers have opted to remain anonymous.

Essential revisions (for the authors):

Both reviewers indicate the interest and importance of the topic, and acknowledge the significant amount of work put in providing evidence for the novel non-canonical Drosha's function mediated by Safb1 in vitro. However, they also raise a number of significant concerns regarding the biological significance of the described observations for the fate determination of adult hippocampal neural stem cells in vivo.

In particular, both reviewers express serious reservations concerning the in vitro cellular systems used to document most of the data reported in the paper.

Moreover, they also indicate that the propose effects on fate determination, as demonstrated now in vitro, are not convincing. The effects on fate determination towards the oligodendrocytic or neuronal fate need to be clearly shown and their in vivo relevance substantiated, and there are also concerns that you do not actually show that Drosha and Safb1 interact directly on the native NFIB's 3'UTR.

In addition to the reviewer's mayor concerns, some of the manuscript sections are difficult to read. In particular, the summary does not completely reflect the content of the manuscript and the introduction section seems to be written in a different level of writing fluidity than the rest of the manuscript, please revise them carefully. In general, It would be advisable to refer consistently to oligodendrocyte differentiation, rather than the more general "glia", and to make a better effort to highlight the novelty and importance of your new findings over your previous work in Rolando et al., 2016.

Please consider carefully the experimental recommendations indicated by the reviewers

*Reviewer #2 (Recommendations for the authors):*

1. The authors should provide stainings of their cultured DG 'NSCs' (under all experimental conditions) with (additional) astrocyte-specific markers (Fabp7 used here also labels astrocytes actually). Especially in Figure 4-Sup.1 the differentiated cultures look astrocyte-like.

2. The astroglial differentiation paradigm illustrated in Figure 7 should be better characterized assessing neuronal/astrocytic/oligodendrocytic marker expression by stainings and/or qPCR.

3. How were the RBPs from the Drosha-interacting protein dataset selected for the gain-of-function experiments in Tet-on DG NSCs? In addition, Sam68 and Tdp4 cannot be found neither in Figure 1G nor in the relevant supplementary table.

4. In most of the IF images of the DG NSCs provided, the +GF ones are acquired at higher magnification than those without GFs, however, scale bars are wrongly indicated in figure legends.

5. The authors should provide immunofluorescence images of the Tet-on 3'UTR in Drosha fl/fl conditions in Figure 2.

6. Are the experiments with the Safb1 mutant construct (lacking the RE-rich region) not performed in primary DG NSCs like all the other experiments, but in N2a cells instead, relevant for NSCs? The limitations of this system should be clearly discussed.

7. In Figure 7E, the Sox10 signal -if anything- increases upon Safb1 OE, in contract to the quantification indicated in Figure 7F. The authors should explain.

8. in vivo staining of Sox10 is mentioned in the text but not shown in Figure 7A.

9. The Sox10 channel should be displayed in white instead of pink in all relevant cases.

*Reviewer #3 (Recommendations for the authors):*

1) As specialized macromolecular complexes involving Safb1 and Drosha might be influenced by presence of additional/cell-specific cofactors, I would suggest repeating some of the key functional experiments in DG NSCs. For example, it is not clear why the modified in vitro processing assay of NFIB HP (shown in Figure 6 F, G) was done with the Drosha/Safb1 complexes precipitated from N2A cells instead from DG NSCs.

2) The authors state that Safb1 KD in cultured DG NSCs led to a rapid increase in activated Casp3 and cell death within 48 hours, preventing further fate analysis. Could be possible to try a different strategy, such as depletion of Safb1 levels or inhibition of its functions? This could be achieved by LNA-antisense oligonucleotides targeting Safb1 transcript, or by overexpression of RNA oligos bearing the NFIB 3'UTR-HP, for example. This strategy should induce oligodendrocyte fate in DG NSCs (in vitro upon GF removal in presence of serum as in Figure 7B), as well as in vivo.

3) Overexpression of Safb1 in V-SVZ NSCs reduced oligodendrocyte differentiation in vitro, is this true also in vivo? What happens to the levels of endogenous NFIB in these cells upon OE of Safb1?

4) Beside Safb1, the expression of NFIB protein in SVZ NSCs and DG NSCs should be also documented in vivo.

---

## [Author Response]

Essential revisions (for the authors):Both reviewers indicate the interest and importance of the topic, and acknowledge the significant amount of work put in providing evidence for the novel non-canonical Drosha's function mediated by Safb1 in vitro. However, they also raise a number of significant concerns regarding the biological significance of the described observations for the fate determination of adult hippocampal neural stem cells in vivo.

We thank the reviewers and editor for their appreciation of our work and the importance of our findings. We deliberated about how to address the concern about the in vivo relevance of our findings. As *Safb1* knockout mice are perinatal lethal, we considered that the best approach to address this major concern was to generate a conditional knockout of *Safb1* (*Safb1* cKO) in adult DG neural stem cells, and address the effects on neural stem cell fate by performing lineage tracing. Therefore, we first had to generate a floxed *Safb1* allele in mice and delete *Safb1* from the adult neural stem cells using *Hes5::CreER^T2^* and to lineage trace the progeny of the *Safb1*-deficient stem cells. This naturally took a considerable amount of time and effort. We present the *Safb1* cKO data in the new Figure 8. We show that *Safb1* cKO from DG neural stem cells of adult mice significantly increases oligodendrocyte production, supporting our in vitro findings. In addition, we could show that loss of SAFB1 results in an increase in expression of the transcription factor NFIB by DG neural stem cells and their progeny supporting that the mechanism of SAFB1 action in controlling stem cell fate in the adult DG is, at least in part, through suppression of NFIB expression. Although this was a major effort, we believe this was the best way to show the relevance of the role of SAFB1 in the DG in vivo.

In particular, both reviewers express serious reservations concerning the in vitro cellular systems used to document most of the data reported in the paper.

The reviewers raised concerns that the cells used in the in vitro assays were not neural stem cells as they had been sorted on the basis of *Hes5::GFP* expression. However, there was a misunderstanding here as the DG neural stem cells were not isolated from the DG by sorting, but were isolated according to the protocol of Babu et al. 2007 (doi: 10.1371/journal.pone.0000388) and cultured and differentiated according to Rolando et al. (doi: 10.1016/j.stem.2016.07.003). We have clarified this in the methods section of the revised manuscript and added a citation to Babu et al.. In addition, we added additional images of the DG neural stem cells and images of the neurons generated after removal of bFGF and EGF (Figure 4—figure supplement 1). In addition, we also stained the cells for “classic” astroglial markers under undifferentiated and differentiated culture conditions. As expected, the DG neural stem cells used do not express GFAP or S100b under undifferentiated culture conditions, but generate a few astrocytes upon differentiation. These data are included in the revised Figure 4—figure supplement 1 in the manuscript.

Moreover, they also indicate that the propose effects on fate determination, as demonstrated now in vitro, are not convincing. The effects on fate determination towards the oligodendrocytic or neuronal fate need to be clearly shown and their in vivo relevance substantiated, and there are also concerns that you do not actually show that Drosha and Safb1 interact directly on the native NFIB's 3'UTR.

We thank the reviewers for this comment. To address this, we have generated a floxed *Safb1* allele and conditionally deleted *Safb1* from adult DG neural stem cells in vivo and followed their fate (new Figure 8). This was a huge effort but we believe that the results convincingly show that SAFB1 regulates oligodendogliogenesis in the adult hippocampus in vivo. In addition, we could show that loss of SAFB1 results in an increase in NFIB protein levels in DG neural stem cells and their progeny. We hope that these new data convince the reviewers of the role of SAFB1 in suppressing oligodendrocyte formation in the adult mouse DG.

…and there are also concerns that you do not actually show that Drosha and Safb1 interact directly on the native NFIB's 3'UTR.

We thank the reviewers for the comment. We had shown with CLIP experiments with anti-SAFB1 antibodies in DG neural stem cells that SAFB1 binds native *Nfib* mRNAs. These results show that endogenous SAFB1 can bind *Nfib* mRNA. These data are also supported by SAFB1 CLIP experiments performed in HepG cells (Van Nostrand et al., 2020). In addition, we showed previously that Drosha interacts directly with native *Nfib* mRNA in DG neural stem cells (Rolando et al. 2016).

The reviewers raised the question of whether Drosha and SAFB1 interact with the same *Nfib* transcript. This was not trivial to address. We needed to develop a novel method of re-RNA precipitation (RIP^2^) of RNA transcripts that simultaneously associate with endogenous Drosha and SAFB1. ReRIP is not an established procedure and, therefore, we developed a method on the basis of a reChIP procedure. These experiments were extremely challenging but clearly show that sequential precipitation of Drosha and then SAFB1 from DG neural stem cells pulls-down the endogenous *Nfib* 3’ UTR region of the RNA transcript (revised Figure 6).

In addition to the reviewer's mayor concerns, some of the manuscript sections are difficult to read. In particular, the summary does not completely reflect the content of the manuscript and the introduction section seems to be written in a different level of writing fluidity than the rest of the manuscript, please revise them carefully. In general, It would be advisable to refer consistently to oligodendrocyte differentiation, rather than the more general "glia", and to make a better effort to highlight the novelty and importance of your new findings over your previous work in Rolando et al., 2016.

We have substantially revised and rewritten the manuscript including the new data and experiments. We have tried to highlight the novelty of our findings showing that SAFB1 controls Drosha activity in adult DG neural stem cell fate decisions. We have completely revised the summary to reflect our findings. We have changed the text to refer to oligodendrocyte formation rather than glial differentiation. We hope that the reviewers appreciate the great effort we have made in addressing their concerns. The reviewer’s comments have certainly improved the manuscript and led to us being able to solidify our interpretations of the results.

Reviewer #2 (Recommendations for the authors):1. The authors should provide stainings of their cultured DG 'NSCs' (under all experimental conditions) with (additional) astrocyte-specific markers (Fabp7 used here also labels astrocytes actually). Especially in Figure 4-Sup.1 the differentiated cultures look astrocyte-like.

We thank the reviewer for their comments and suggestions. We now provide additional images of immunostains of the DG neural stem cells with astrocytic markers (GFAP and S100b: Figure 4—figure supplement 1). We have also provided more representative images of the cells in the revised Figure 4—figure supplement 1.

2. The astroglial differentiation paradigm illustrated in Figure 7 should be better characterized assessing neuronal/astrocytic/oligodendrocytic marker expression by stainings and/or qPCR.

We thank the reviewer for the suggestion. The differentiation protocol we used to induce gliogenesis are based on previous experiments and are described in Rolando et al. 2016. However, we have added additional images of immunostains of the DG neural stem cells with astrocytic markers (GFAP and S100b) under undifferentiating and differentiating conditions.

3. How were the RBPs from the Drosha-interacting protein dataset selected for the gain-of-function experiments in Tet-on DG NSCs? In addition, Sam68 and Tdp4 cannot be found neither in Figure 1G nor in the relevant supplementary table.

The RBPs we show were selected based on their interaction with Drosha and NFIB RNA HP regions and ranking across the various assays. However, we chose those that were pulled-down with Drosha from DG neural stem cells. The lists we present in the excel sheets contain the genbank information which includes the gene ID and gene name. As some proteins have different gene names, we considered it was more consistent to use the gene names. The gene encoding SAM68 is *Khdrbs1* which can be found in multiple lists in the data sheets. The gene encoding TDP43 is *Tardbp*, which is also present in many of the data sets presented in Table S2. In Figure 1G, Tardbp is present in the RBPs enriched in the *Nfib* 5’ UTR HP pull-down list (list C in Table S3 iVenn).

4. In most of the IF images of the DG NSCs provided, the +GF ones are acquired at higher magnification than those without GFs, however, scale bars are wrongly indicated in figure legends.

We have checked and confirmed the magnifications and scale bars in all of the figures.

5. The authors should provide immunofluorescence images of the Tet-on 3'UTR in Drosha fl/fl conditions in Figure 2.

We have now added images of the Tet-on 3’UTR cells to the revised Figure 2.

6. Are the experiments with the Safb1 mutant construct (lacking the RE-rich region) not performed in primary DG NSCs like all the other experiments, but in N2a cells instead, relevant for NSCs? The limitations of this system should be clearly discussed.

We thank the reviewer for their comments. We turned to a heterologous cell type N2A for these biochemical experiments as N2A cells are considerably easier to transfect than primary DG neural stem cells. Therefore, in order to have consistent and sufficient material, a cell line that expresses Drosha, SAFB1 and NFIB mRNA was needed for these experiments. It should be noted that neuroblastoma N2A cells express endogenously Drosha, SAFB1 and *Nfib* mRNA.

7. In Figure 7E, the Sox10 signal -if anything- increases upon Safb1 OE, in contract to the quantification indicated in Figure 7F. The authors should explain.

Unfortunately, we are not sure what the reviewer is referring to here. The quantification shown in Figure 7E shows the quantification of levels of SAFB1 protein in DG versus V-SVZ neural stem cells (mean intensity). In Figure 7G we show the effects of SAFB1 OE on Sox10 expression. In this experiment, the SAFB1 overexpressing cells (lower panels) are labelled with CFP and indicated with arrows. In Figure 7H, we show the quantification of proportion of transfected cells (CFP+) that express Sox10 (Sox10^+^CFP^+^) and not the levels. We hope this clarifies the issue for the reviewer.

8. in vivo staining of Sox10 is mentioned in the text but not shown in Figure 7A.

We thank the reviewer for their comment. We have included immunostains for SAFB1 and Sox10 expression in the V-SVZ in vivo in Figure 7B and the new Figure 8 during the analysis of the *Safb1* cKO mice.

9. The Sox10 channel should be displayed in white instead of pink in all relevant cases.

We thank the reviewer for this suggestion and have changed the figures accordingly.

Reviewer #3 (Recommendations for the authors):1) As specialized macromolecular complexes involving Safb1 and Drosha might be influenced by presence of additional/cell-specific cofactors, I would suggest repeating some of the key functional experiments in DG NSCs. For example, it is not clear why the modified in vitro processing assay of NFIB HP (shown in Figure 6 F, G) was done with the Drosha/Safb1 complexes precipitated from N2A cells instead from DG NSCs.

We thank the reviewer for the comment. The reason we had to use N2A cells for isolation of the SAFB1-Drosha complexes was that knockdown of *Safb1* mRNA in DG neural stem cells in vitro resulted in cell death under undifferentiating culture conditions – this we stated in the manuscript. Hence, in order to generate endogenous SAFB1-Drosha complexes where SAFB1 levels were reduced in a controlled manner by knockdown, we had to turn to an alternative cell type. As N2A cells express SAFB1Drosha and *Nfib* mRNA, and Drosha seems to regulate NFIB protein levels in these cells, we considered that the SAFB1-Drosha complexes in N2A play similar roles in N2A as in DG neural stem cells.

2) The authors state that Safb1 KD in cultured DG NSCs led to a rapid increase in activated Casp3 and cell death within 48 hours, preventing further fate analysis. Could be possible to try a different strategy, such as depletion of Safb1 levels or inhibition of its functions? This could be achieved by LNA-antisense oligonucleotides targeting Safb1 transcript, or by overexpression of RNA oligos bearing the NFIB 3'UTR-HP, for example. This strategy should induce oligodendrocyte fate in DG NSCs (in vitro upon GF removal in presence of serum as in Figure 7B), as well as in vivo.

We thank the reviewer for the suggestions. Indeed, we used antisense esiRNA knockdown in DG neural stem cells in an attempt to reduce SAFB1 levels and this induced cell death. Overexpressing NFIB 3’ UTR hairpin is an interesting idea. However, when we performed the activity screening experiments with the Tet-regulated *Nfib* hairpin containing expression vectors (where the 3’ UTR hairpin is highly expressed) we did not see effects on stem cell survival or differentiation. This we interpret to suggest that the Drosha cleavage activity on *Nfib* RNA is highly efficient (which we also show in the manuscript and in Rolando et al. 2016) and, hence, it is not easy to saturate the process under physiological conditions.

3) Overexpression of Safb1 in V-SVZ NSCs reduced oligodendrocyte differentiation in vitro, is this true also in vivo? What happens to the levels of endogenous NFIB in these cells upon OE of Safb1?

We thank the reviewer for this important question. We have not performed over expression of SAFB1 in vivo. In the revisions, we focused on the essential requirements and addressed the knockout of *Safb1* and the co-binding of SAFBI and Drosha to endogenous NFIB. Although this is a very important question, we focused on addressing the role of SAFB1 in the DG neural stem cells during the revisions. We are planning to address the roles of SAFB1 in SVZ and other neural stem cells in the future.

4) Beside Safb1, the expression of NFIB protein in SVZ NSCs and DG NSCs should be also documented in vivo.

In the revised manuscript we now document and quantified NFIB levels in DG neural stem cells and show images before and after *Safb1* ablation (Figure 8 E and F). We include Author response image 1 showing NFIB expression in the V-SVZ and adjacent striatum. As the manuscript is focused on the role of SAFB1 in the DG and not the in adult SVZ, we believe this data does not add important information to the manuscript. We are addressing additional roles of SAFB1 in neurogenesis including in the V-SVZ, but these analyses are beyond the scope of this manuscript.

**Author response image 1. sa2fig1:** Expression of NFIB in the V-SVZ of adult Hes5::CreER^T2^ Rosa26-CAG::Egfp mice 21 days after Tamoxifen induction (as described in Figure 8A of the manuscript) and lineage tracing of V-SVZ neural stem cells and their progeny (GFP^+^). NFIB expression was found in the stem and progenitor cells of the V-SVZ (arrowheads) and GFAP^+^ astrocytes in the striatum (*).